# Richness and biogeography of Pteridoflora in montane forests of eastern Mexico

**J. Daniel Tejero-Díez**[1], **Perla V. Rodríguez-Sánchez**[1], **Raúl Contreras-Medina**[2],
**Julio Cesar Ramírez-Martínez**[3], **Jonathan D. Amith**[4], **Leccinum J. García-Morales**[5],
**Celia Sanginés-Franco**[3], **Isolda Luna-Vega**[3]*

1 Facultad de Estudios Superiores Iztacala, Laboratorio de Botánica Estructural, Carrera de Biología, Universidad Nacional Autónoma de México (UNAM), Tlalnepantla, Estado de México, México, 2 Facultad de Ciencias, Laboratorio de Biodiversidad, Universidad Autónoma Benito Juárez de Oaxaca (UABJO), Oaxaca de Juárez, Oaxaca, México, 3 Departamento de Biología Evolutiva, Facultad de Ciencias, Laboratorio de Biogeografía y Sistemática, Universidad Nacional Autónoma de México (UNAM), México City, México, 4 Department of Botany, National Museum of Natural History, Smithsonian Institution, Washington DC, United States of America, 5 Herbario, División de Estudios de Posgrado e Investigación, Instituto Tecnológico de Ciudad Victoria (ITCV), Ciudad Victoria, Tamaulipas, México

* luna.isolda@gmail.com

## Abstract

This study compiles and updates the checklist of ferns and lycophytes from the Sierra Madre Oriental (SMOR), Mexico. We analyzed the distribution and richness of these groups of seedless vascular plants to ensure the accuracy and reliability of the data. We reviewed information on these taxa from regional floristic studies, field explorations, and herbarium specimens. Our updated list includes 567 species, of which 511 are ferns and 56 lycophytes, organized into 37 families and 137 genera. The five most species-rich families are Pteridaceae (131), Polypodiaceae (76), Dryopteridaceae (68), Aspleniaceae (45), and Selaginellaceae (41). The five most species-rich genera are *Asplenium* (42), *Elaphoglossum* (30), *Pleopeltis* (24), *Myriopteris* (22), and *Selaginella* (21). The Mexican states with the greatest species richness within the SMOR are Puebla (423), Hidalgo (322), Querétaro (227), and Tamaulipas (181). The five grid cells with the highest richness, each containing more than 190 species, are situated in the southern part of the SMOR (Puebla and adjacent parts of Veracruz). The physiographic provinces with the greatest diversity are Carso Huasteco and the Gran Sierra Plegada, with 523 and 175 species, respectively. Latitudinal analysis indicated low richness at the northern extreme (28 °- 29 ° N, 11 species) and high richness at the southern extreme (19 °- 20 ° N, 469 species), with a notable peak at 24 °- 25 ° N (350 species). The highest diversity within elevation intervals occurs at 1,000–1,500 m (455 species). The SMOR contains 22 species of the pteridoflora classified under various risk categories according to the Mexican Official Norm NOM-059-SEMARNAT-2010, including seven listed on the IUCN Red List and ten protected under CITES. The taxonomic diversity index shows that the SMOR is among Mexico's most species-rich mountain ranges. The species richness

**Data availability statement:** All relevant data are within the manuscript and its Supporting Information files.

**Funding:** This research was supported by Project DGAPA-PAPIIT IN219424 and SECIHTI CB-2025-G-499.

**Competing interests:** The authors have declared that no competing interests exist.

documented here in the SMOR accounts for 50–54% of the pteridophyte diversity recorded in Mexico, increasing to 59.3% based on recent floristic studies conducted in the region.

## Introduction

In the montane areas of tropical regions, the lycophytes and ferns are two lineages of vascular plants that are highly diverse worldwide [1–3]. These two seedless plant groups are known as pteridobionts because they share the same double-phase life cycle [4–6]. This taxonomic success and abundance in environments with high atmospheric humidity is due to both the sexual phase (zygote formation) and the sporophyte phase (low stomatal reactivity) being sensitive to water deficit [7–9]. These characteristics make them ideal indicators of the conservation quality of tropical montane cloud forests [10,11]. Therefore, biological studies in biogeography, ecology, or taxonomy are crucial to our understanding of various aspects of ecosystem management.

In the Neotropics, Mexico is one of the countries with a highly diverse pteridoflora [1] according to Hassler [12] with 1,057 species, including 99 endemics. If hybrids and infraspecies are included, the number of taxa increases to 1,123 (sensu Tejero-Díez database). These taxa account for almost 5% of the country's total vascular flora, which includes 23,314 species (sensu [13]), of which 20% are endemic [14]. The Mexican pteridoflora has been extensively studied from a taxonomic perspective [15], and recent nomenclatural updates have been incorporated into the Mexican checklists of ferns and lycophytes [12]. Nonetheless, investigating certain regions —where information remains scattered or incomplete— is still necessary. Ultimately, understanding the historical processes behind distribution and the ecological and environmental relationships within species is crucial.

Knowledge about the spatial patterns of richness and endemism in Mexican pteridoflora is limited, especially in highly biodiverse areas. The Sierra Madre Oriental (SMOR) is the second largest mountain range and is regarded as a highly biodiverse Mexican biogeographic province [16–18], where ferns and lycophytes contribute significantly to this biodiversity. The SMOR boasts a higher floristic richness than any other mountainous region in Mexico, with 8,472 species [18]. This richness is attributed to several factors, such as its geographic location on both sides of the Cancer tropic, complex topography, diverse climate types, the presence of nearly all Mexican vegetation types, and a broad elevational gradient ranging from 100 to over 3,600 meters [17].

Previous floristic checklists and biogeographic research related to ferns and lycophytes have been conducted in the main mountain ranges of Mexico, such as the Sierra Madre del Sur [6], the Sierra Madre Oriental [19,20], the Trans-Mexican Volcanic Belt [21], and the Sierra Madre Occidental [14], as well as the Chiapas Highlands [22]. Recently, floristic checklists of the SMOR covering all vascular plants have been published [18,19], but our review of the pteridoflora has identified issues

like synonymies, incorrect geographic coordinates, mistaken taxonomic identifications or confusion of particular species, as well as species not recorded in the cited studies but present in the SMOR. Additionally, the study by [20], conducted in the SMOR, includes only a subset of leptosporangiate ferns, the most diverse group of ferns, with an estimated 10,323 species worldwide [23].

The aims of this study were: (1) to carry out an updated and revised floristic inventory of the pteridoflora of the SMOR and (2) to analyze the distributional patterns of ferns and lycophytes using three different area units (states, physiographic provinces and grid cells of 30 × 30 minutes latitude/longitude) and to identify elevational and latitudinal richness patterns, (3) to compare the species richness of the SMOR with other Mexican mountain chains and among the physiographic provinces of the SMOR using a taxonomic diversity index, and (4) to revise the categories of threatened taxa of ferns and lycophytes based on the IUCN Red Lists, the Convention on International Trade in Endangered Species of Wild Fauna and Flora (CITES), and the federal regulatory framework established by the Mexican government (NOM-059-SEMARNAT-2010).

## Materials and methods

### Study area

The Sierra Madre Oriental (SMOR) is located in northeastern Mexico (Fig 1A) and is the country's second most important and extensive mountainous province system [16,24]. This montane region is regarded as part of the Mesoamerican hotspot proposed for ferns and lycophytes by [3]. Geologically, the SMOR resulted from the deformation of Cenozoic and Mesozoic marine sedimentary rocks that were uplifted, shortened, and transported north-eastward, creating a fold-and-thrust belt during the Laramide orogeny [25].

The SMOR extends 800 km in length, with an amplitude of 80–100 km [25]. Although various biotic and abiotic criteria have led to different proposed boundaries for the SMOR [16], we adopted the regionalisation framework of [26], which is based on physiographic regionalisation (Fig 1A). This regionalisation scheme of the SMOR has also been applied in studies examining the distribution of Cactaceae [27], leptosporangiate ferns [20], and vascular plants [18]. Within this mountain system, 10 physiographic subprovinces are recognized: Sierras Transversales, Pliegues de Saltillo-Parras, Sierras y Llanuras Occidentales, Gran Sierra Plegada, Sierra de San Carlos, Sierra de Tamaulipas, Carso Huasteco, Serranía de El Burro, Sierra de la Paila, and Sierras y Llanuras Coahuilenses. The SMOR constitutes one biogeographic province of Mexico [16,24], and together with the Trans-Mexican Volcanic Belt, the Sierra Madre del Sur, and the Sierra Madre Occidental, forms part of the Mexican Transition Zone [24]. The SMOR covers parts of the Mexican states of Coahuila, Durango, Guanajuato, Hidalgo, Nuevo León, Puebla, Querétaro, San Luis Potosí, Tamaulipas, Veracruz, and Zacatecas (Figs 1B, C). Its latitudinal extent spans from the 20th to the 30th parallel, bounded to the north by the Grande (Bravo) River and to the south by the Trans-Mexican Volcanic Belt [16]. Its location places it in contact with key atmospheric systems, including the subtropical high-pressure belt, trade winds, winter cold fronts, and summer tropical storms linked to hurricanes. This, combined with a latitudinal range of ten degrees, and elevational variations from around 100 m in the lowlands of Tamaulipas and San Luis Potosí [17] to 3,713 m above sea level at Cerro El Potosí in Nuevo León (with an average of 1,000–1,500 m, Fig 1C), and the mountain chain running parallel to the Gulf of Mexico opposite the entry of humid winds, produces a significant orographic shadow effect. This results in a complex climate mosaic, with sites experiencing high rainfall, others experiencing dry seasons, and warm- and temperate-climate conditions, and gradients between these conditions [28]. Furthermore, there are isothermal regimes and extremes, with temperature fluctuations of up to 14°C between the hottest and coldest months. Generally, the climate becomes drier and more extreme towards the north of the SMOR, with its western slope facing the Mexican Plateau. Nearly all recognized Mexican climatic types are present within this area. Vegetation in the SMOR includes xeric shrubland, conifer forests, oak forests, montane cloud forests, tropical forests, and mixed forests [16,17] (Fig 1B).

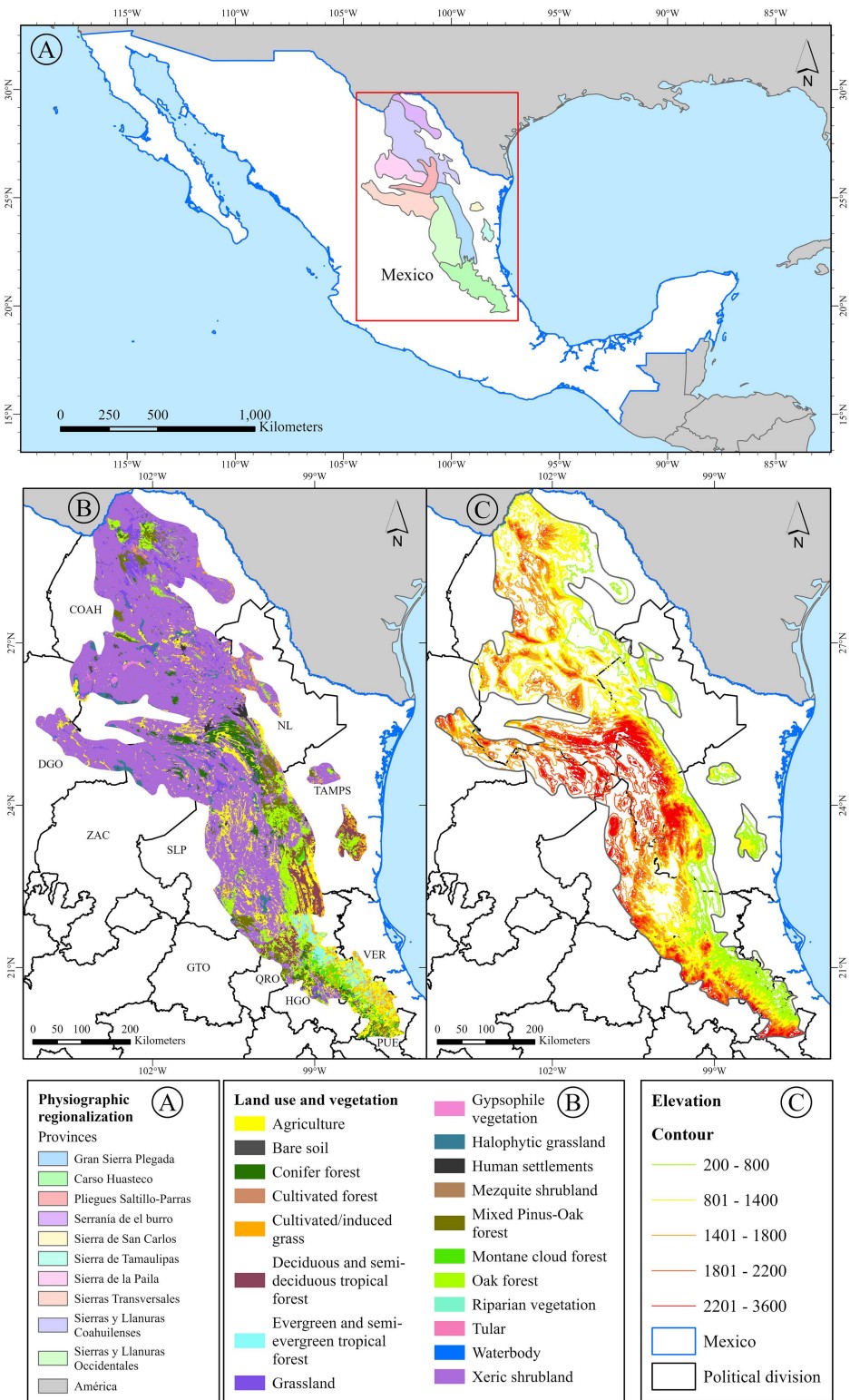

**Fig 1. Maps of northeastern Mexico covering the Sierra Madre Oriental (SMOR). A.** Physiographic regionalization. **B.** Land use and vegetation types. **C.** Elevation. State abbreviations: Coahuila = COAH, Durango = DGO, Nuevo León = NL, Zacatecas = ZAC, Tamaulipas = TAMPS, Querétaro = QRO,

Veracruz = VER, Guanajuato = GTO, Puebla = PUE, San Luis Potosí = SLP, Hidalgo = HGO. Map created by the authors using ArcGIS Pro. All vector layers (Physiography, Land Use and Vegetation Series VII, Hypsometry/Topography, and Political Division) were sourced from INEGI (https://www.inegi.org.mx/), used under the Terms of Free Use of Information which permit commercial utilization.

## Data sources

We obtained distributional data for ferns and lycophytes from three main sources: 1) the review of floristic studies conducted in the SMOR [4,18,19,29–61], 2) the revision of herbarium vouchers deposited in herbaria such as ARIZ, ASU, BH, BM, BRIT, BRY, CIB, CNS, DUKE, ENCB, F, FCME, GBH, HGOM, HPC, HUAP, HZAC, IEB, IND, ITCV, LAM, LL, MEXU, MICH, MIN, MMNS, MO, MSC, MU, NCU, NY, P, PH, QMEX, SJNM, TAES, TEX, UAMIZ, US, UAT, UC, UCR-BPS, USF, UT, UTEP, VDB, WIS, WLM, WTU, XAL, and YU (acronyms following [62], 3) field surveys in some Mexican states, such as Hidalgo, Querétaro, Puebla, and Tamaulipas, to collect botanical specimens. Voucher specimens collected during the fieldwork were deposited at the National Herbarium of Mexico (MEXU), herbarium of the Benemérita Universidad Autónoma de Puebla (HUAP), the herbarium of the Instituto Tecnológico de Ciudad Victoria (ITCV), and the United States National Herbarium (US). We revised all taxonomic names and excluded synonyms and misidentifications, as described in [12]. We reviewed the known distribution as described in [15]. In this study, we excluded *Microsorum grossum* (Langsd. & Fisch.) S.B. Andrews, an exotic species growing as an epiphyte in nearby zones of Gómez Farías, Tamaulipas. At the final stage of this research, *Adiantum wilesianum* Hook. was recorded in three localities, two from San Luis Potosí and one from Hidalgo, which are not represented in our analyses.

We verified the accurate identification of endemic species and those considered rare (with 1–3 specimens in the total number of herbaria reviewed or field collections). The accurate identification of names that did not match expert reports in the literature was also reviewed. We compiled a database from all these information sources, including 4,479 georeferenced records of lycophytes and ferns in the SMOR. Using this information, we created geographic distribution maps of the species with ArcGIS Pro [63].

Since the publication of 'The Pteridophytes of Mexico' by [15], new combinations, genera, species, and records of ferns and lycophytes have been published and reported. For these reasons, and in accordance with the International Code of Nomenclature for algae, fungi, and plants (Shenzhen Code) [64], we verified the correct names of each species. The nomenclature and author citations were standardized following the [12] database (except in *Selaginella* sensu [65] and updated to align with the latest integrative and circumscription taxa systematics based on the proposal of the Pteridophyte Phylogeny Group [23].In our database, as is common with herbarium specimens, duplicate samples are found across different botanical collections. In our study, we removed duplicate records from the biogeographic analysis to ensure unique locations. Additionally, we identified and corrected records with swapped latitude and longitude values. We also excluded specimens with vague localities and inconsistencies. For herbarium specimens lacking geographic coordinates, we used topographic maps at a 1:50,000 scale from the Instituto Nacional de Estadística y Geografía (INEGI) and the previously published locality list for the Sierra Madre Oriental [66] to obtain the corresponding coordinates. Ultimately, we aimed to include only georeferenced and verified data in our study.

To visualize the spatial distribution of collection records and identify potential sampling biases across the SMOR, we created a heatmap from all georeferenced occurrences included in the database. The heatmap was generated in ArcGIS Pro [63] using the Point Density tool, which measures the spatial intensity of records. This method allowed us to highlight areas with high densities of fern and lycophyte collections, enabling the detection of well-sampled regions and zones with limited survey effort. The resulting layer served as a valuable tool for evaluating the dataset's spatial completeness and providing context for subsequent analyses of species richness and distribution patterns.

## Biogeographic and diversity analyses

Mexican states included in the SMOR, physiographic provinces were used to perform the biodiversity analyses, and 30 × 30 minute grid cells were used as units of analysis in ArcGIS Pro [63]. In the grid analysis, we recorded 86 grid cells containing at least one fern or lycophyte species from this framework. The spatial size of grid cells provides a reliable spatial resolution for the distribution compared with other studies using plants in the SMOR [18,20,27] regarding biodiversity data and biological conservation. We selected this scale size because similar scales have been tested in previous studies on biogeography and the diversity of different biological groups of the Mexican biota [67,68], including ferns and lycophytes [69].

The total expected richness of ferns and lycophytes in the SMOR was estimated using a nonparametric abundance model (Chao1) in EstimateS [70] to evaluate its performance and determine whether it fits the data.

In the grid cell analysis, we used three standard biodiversity metrics (species richness, weighted endemism, and corrected weighted endemism) to measure the biodiversity patterns previously used for different groups of the Mexican pteridoflora [20,69,71,72]. These metrics are more than just theoretical concepts; they are practical tools that have been effectively used in the study of the Mexican pteridoflora [73]. We measured species richness as the total number of species present within each grid cell. Each fern or lycophyte species was counted as present in a grid cell, regardless of whether it was recorded once or multiple times in that cell [27,74]. The weighted endemism (WE) involves several steps [73]. First, we divided each grid cell occurrence by the number of grid cells in which one species occurs. For example, a fern species limited to a single grid cell was scored as '1' for that grid cell and '0' for all remaining grid cells, whereas a fern species found in five grid cells was scored as '0.2' for each of the five grid cells and '0' for all other grid cells; then we obtained the sum of all score species values for each grid cell. We calculated the corrected weighted endemism (CWE) by dividing the WE value by the number of species in a grid cell [73,74]. The WE index was considered a biodiversity measure sensitive to species richness, suggesting that richness patterns partially mirror the WE value [27,74]. In contrast, the CWE index emphasizes grid cells that contain a proportion of species with limited ranges, which are not necessarily linked to species-rich grid cells [27,74].

The biodiversity metrics mentioned above were calculated using the software Biodiverse, version 3.0 [75], and mapped in ArcGIS Pro [63]. Grid cells containing only one species were excluded from the CWE analysis because they lack overlapping distributions, which is a key criterion for identifying areas of endemism [76,77]. Grid cells with the highest scores in the WE index were considered centers of richness, while those with the highest values in the CWE index were regarded as centers of endemism [73,74].

The geographic distribution of ferns and lycophytes inhabiting the SMOR was established based on the study of [15] as a general framework. We defined nine categories related to the geographic distribution of each species as follows: 1) Cosmopolitan, taxa distributed worldwide; 2) American, taxa distributed across the American continent; 3) Nearctic, taxa found in Mexico (and sometimes in Mesoamerica) and North America; 4) Neotropical, taxa found in Mexico, Central America, the Antilles, and South America; 5) Mesoamerican, taxa distributed in Mexico and Central America; 6) Endemic to the Mexican territory; 7) Endemic to the SMOR, taxa with a known distribution limited to the boundaries of the SMOR; 8) Regional endemic, in adjacent areas to the SMOR; and 9) Introduced taxa.

We determined the frequency of each fern and lycophyte species following the method recommended by [78], based on the number of records for each species and assigned them to their respective frequency classes. This reflects that some species will thrive at the expense of others, leading to differences in species frequency, with a few species having many individuals and others being rare. We divided the species into five classes of record frequency according to the following ranges: A, 1–11 records; B, 12–23; C, 24–35; D, 36–47; and E, 48–59.

Typically, the species richness of a specific flora is expressed as the number of species or as a biodiversity index that accounts for the number of species per area on a logarithmic scale. To compare the richness of lycophytes and ferns among the main Mexican mountain ranges and the physiographic provinces of the SMOR, we used the taxonomic

diversity index (IB), which accounts for the number of species per area on a logarithmic scale, following the formula IB = S/Ln A, where S is the number of recorded species and A is the size of the area [79]. Therefore, using biodiversity indices is crucial for comparing sampled areas of different sizes. Using this formula, we assessed species richness across geographic regions, specifically the main Mexican mountain ranges and the physiographic provinces of the SMOR.

Finally, we assessed the species richness of lycophytes and ferns across different latitudes and elevation gradients. The SMOR covers a wide elevation range from 100 m to 3,713 m above sea level. Based on this, we divided the elevation into eight intervals of 500 m each to illustrate species distribution along this gradient in a graph. These intervals have been used in previous studies on ferns and lycophytes in other Neotropical areas [14,80–82], so we applied them to our data for comparison.

We divided the SMOR into 1° latitudinal belts to assess the relationship between species richness and latitude. Additionally, species richness was calculated for all grid cells within each 1° latitude belt from 19°-20° N to 28°-29° N. The species richness values for each latitudinal belt were determined using the taxonomic diversity index (IB) for that belt [83,84]. We presented the results of this analysis on a graph showing the taxonomic diversity index values against latitude.

## Conservation status

We revised the threatened taxa categories for ferns and lycophytes inhabiting the SMOR in accordance with the Mexican federal regulatory framework [85], the IUCN Red Lists [86], and the Convention on International Trade in Endangered Species of Wild Fauna and Flora (CITES) [87]. Regarding the Mexican risk categories, we reviewed the Mexican Official Norm NOM-059-SEMARNAT-2010 [85] (hereafter, NOM-059) to determine how many ferns or lycophytes are listed and to confirm the assigned risk category for each species. The NOM-059 is the official document issued by the Mexican government, and lists threatened species of plants, fungi, and animals [85]. This document outlines the federal regulatory framework, including the specifications for inclusion and analysis to assess the risk of threatened species. Similarly, the risk categories were assessed according to the IUCN Red List of Threatened Species [86] and the CITES [87] lists. Comparing the IUCN Red List with the Mexican Official Norm NOM-059, some categories can be considered equivalent, such as the threatened category in the Mexican Official Norm to the vulnerable category of the IUCN. In contrast, the special protection category of the NOM-059 includes some minor categories from the IUCN [88].

## Results

Based on examined herbarium specimens, web databases, specialized literature, and specimens collected during field expeditions, we compiled a database of 4,479 georeferenced accession records (Fig 2A). These records correspond to 567 species from the SMOR, belonging to 137 genera and 37 families (S1 Appendix). From this checklist, 511 species (34 families, 124 genera) are from Polypodiidae, while 56 (3 families, 13 genera) belong to Lycopodiidae. We identified 211 species of ferns and lycophytes that were not previously documented in the floristic studies conducted by [19] and [68] in the SMOR. Of the total names referenced in the studies, 60 were rejected or excluded because they were synonyms or did not represent species found in the study area or the country (S1 Appendix).

The heatmap (Fig 2B) visually shows the record density and reveals spatial patterns in sampling effort. It displays the distribution of collection records, with yellow and red colours indicating areas with a high number of records, and blue indicating areas with fewer records, interpreted as well-sampled and under-sampled regions, respectively. The highest concentration of records is located in the southern portion of the SMOR in the states of Hidalgo, Puebla, Querétaro, and Veracruz.

We found that the most diverse families (20 or more species) in the SMOR are Pteridaceae (131 species; 23.1%), Polypodiaceae (76; 13.4%), Dryopteridaceae (68; 12.0%), Aspleniaceae (45; 7.9%), Selaginellaceae (41; 7.2%), Thelypteridaceae (34; 6.0%), and Hymenophyllaceae (28; 4.9%) (Table 1). Nine families are represented by a single species

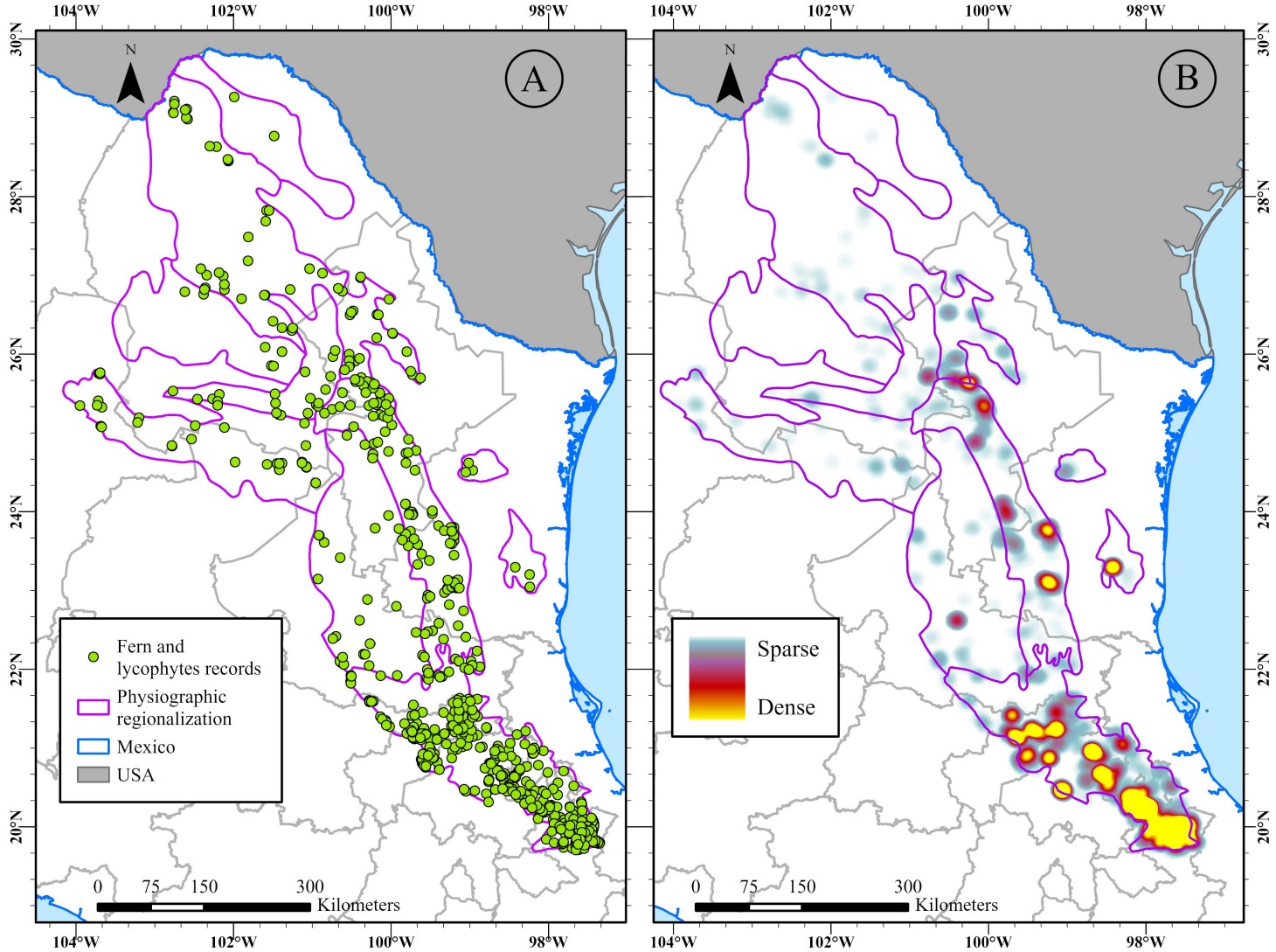

**Fig 2. A. Map of the Sierra Madre Oriental (SMOR) showing georeferenced records of ferns and lycophytes. B.** Heatmap showing the sampling effort of ferns and lycophytes. Map created by the authors using ArcGIS Pro. Physiographic regionalization (SMOR) and Political Division layers were sourced from INEGI (https://www.inegi.org.mx/), used under the Terms of Free Use of Information which permit commercial utilization.

each: Culcitaceae, Didymochlaenaceae, Lindsaeaceae, Lonchitidaceae, Lygodiaceae, Onocleaceae, Plagiogyriaceae, Saccolomataceae, and Schizaeaceae.

## State diversity in the SMOR

Our analysis indicates that Puebla has the highest number of fern and lycophyte species, with 423 species (31.9 species/km$^2$), followed by Hidalgo and Querétaro, which have 321 (23.3 species/km$^2$) and 227 (17.6 species/km$^2$), respectively. Tamaulipas and Veracruz have 181 (9.8 species/km$^2$) and 117 (8.3 species/km$^2$), respectively, while the other states of the SMOR have fewer than 100 species each. Endemic ferns or lycophytes confined to a single Mexican state were recorded in Puebla and Tamaulipas, with three taxa, and in Nuevo León and San Luis Potosí, with two species.

**Table 1. Families of ferns and lycophytes represented in the Sierra Madre Oriental. We recorded the number of species and genera in each family and endemics.**

| Families | Species (%) | Genera | Endemic species | Most diverse genera |
|---|---|---|---|---|
| Anemiaceae | 12 (2.1) | 1 | *Anemia* 1 | |
| Aspleniaceae | 45 (7.9) | 2 | *Asplenium* 6 | *Asplenium* (42) |
| Athyriaceae | 19 (3.4) | 3 | | |
| Blechnaceae | 12 (2.1) | 5 | *Woodwardia* 1 | |
| Culcitaceae | 1 (0.2) | 1 | | |
| Cyatheaceae | 10 (1.8) | 3 | | |
| Cystopteridaceae | 2 (0.4) | 1 | | |
| Dennstaedtiaceae | 13 (2.3) | 6 | | |
| Dicksoniaceae | 2 (0.4) | 2 | | |
| Didymochlaenaceae | 1 (0.2) | 1 | | |
| Dryopteridaceae | 68 (12.0) | 12 | *Ctenitis* 2, *Dryopteris* 1, *Elaphoglossum* 10, *Megalastrum* 2, *Phanerophlebia* 1 | *Elaphoglossum* (30) |
| Equisetaceae | 4 (0.7) | 1 | | |
| Gleicheniaceae | 7 (1.2) | 4 | | |
| Hymenophyllaceae | 28 (4.9) | **7** | *Didymoglossum* 1 | |
| Isoetaceae | 2 (0.4) | 1 | *Isoetes* 2 | |
| Lindsaeaceae | 1 (0.2) | 1 | | |
| Lomariopsidaceae | 2 (0.4) | 1 | *Lomariopsis* 1 | |
| Lonchitidaceae | 1 (0.2) | 1 | | |
| Lycopodiaceae | 13 (2.3) | 6 | *Huperzia* 1 | |
| Lygodiaceae | 1 (0.2) | 1 | | |
| Marattiaceae | 4 (0.7) | 2 | | |
| Marsileaceae | 3 (0.5) | 1 | | |
| Nephrolepidaceae | 6 (1.1) | 1 | | |
| Onocleaceae | 1 (0.2) | 1 | | |
| Ophioglossaceae | 11 (1.9) | 5 | | |
| Osmundaceae | 2 (0.4) | 2 | | |
| Plagiogyriaceae | 1 (0.2) | 1 | | |
| Polypodiaceae | 76 (13.4) | 18 | *Campyloneurum* 1, *Pecluma* 1, *Pleopeltis* 10, *Polypodium* 3 | *Pleopeltis* (24) |
| Psilotaceae | 3 (0.5) | 1 | | |
| Pteridaceae | 131 (23.1) | 26 | *Adiantum* 1, *Argyrochosma* 2, *Aspidotis* 1, *Astrolepis* 1, *Bommeria* 2, *Gaga* 5, *Myriopteris* 4, *Notholaena* 6, Pellaea 2, Ynesmexia 1 | *Myriopteris* (22) |
| Saccolomataceae | 1 (0.2) | 1 | | |
| Salviniaceae | 2 (0.4) | 1 | | |
| Schizaeaceae | 1 (0.2) | 1 | | |
| Selaginellaceae | 41 (7.2) | 6 | *Bryodesma* 4, *Lepidoselaginella* 2, *Pulviniella* 1, *Selaginella* 7 | *Selaginella* (21) |
| Tectariaceae | 3 (0.5) | 1 | | |
| Thelypteridaceae | 34 (6.0) | 8 | *Amauropelta* 1, *Goniopteris* 1, *Pelazoneuron* 2 | |
| Woodsiaceae | 3 (0.5) | 1 | *Physematium* 1 | |
| **TOTAL 37** | **567** (100.0) | **137** | **89** | **139** |

From the total identified pteridoflora of the SMOR area as defined by [26], we identified an essential subset of 123 species representing new records for the SMOR. A subset of 40 species has an updated distributional range, previously recorded in the SMOR, and is now documented in other states not previously reported in the literature (Fig 3).

Eleven of the new records belong to *Elaphoglossum*, seven to *Selaginella*, and six are filmy ferns of the genus *Hymenophyllum*. A total of 41 genera are represented by a single species within the new records in the SMOR. The others have between two and five species. The genera with the most species showing range extension into the SMOR are *Notholaena* and *Amauropelta*, with five and four taxa, respectively. Overall, 16 genera have one species extending its range in the SMOR, while the rest are represented by two or three species.

**Common vs. rare species**

As is common in biogeographic studies based on herbarium data, few fern species are represented by a disproportionate number of specimens, while many are represented by only a few (Fig 4). In our study, only four species have more than 50 records (*Phlebodium pseudoaureum*, *Pleopeltis polylepis*, *Pleopeltis plebeia*, and *Polypodium plesiosorum*), which contrasts with several other species limited to just one or three localities (252 taxa). These latter data account for approximately 50% of the pteridoflora of this mountain range.

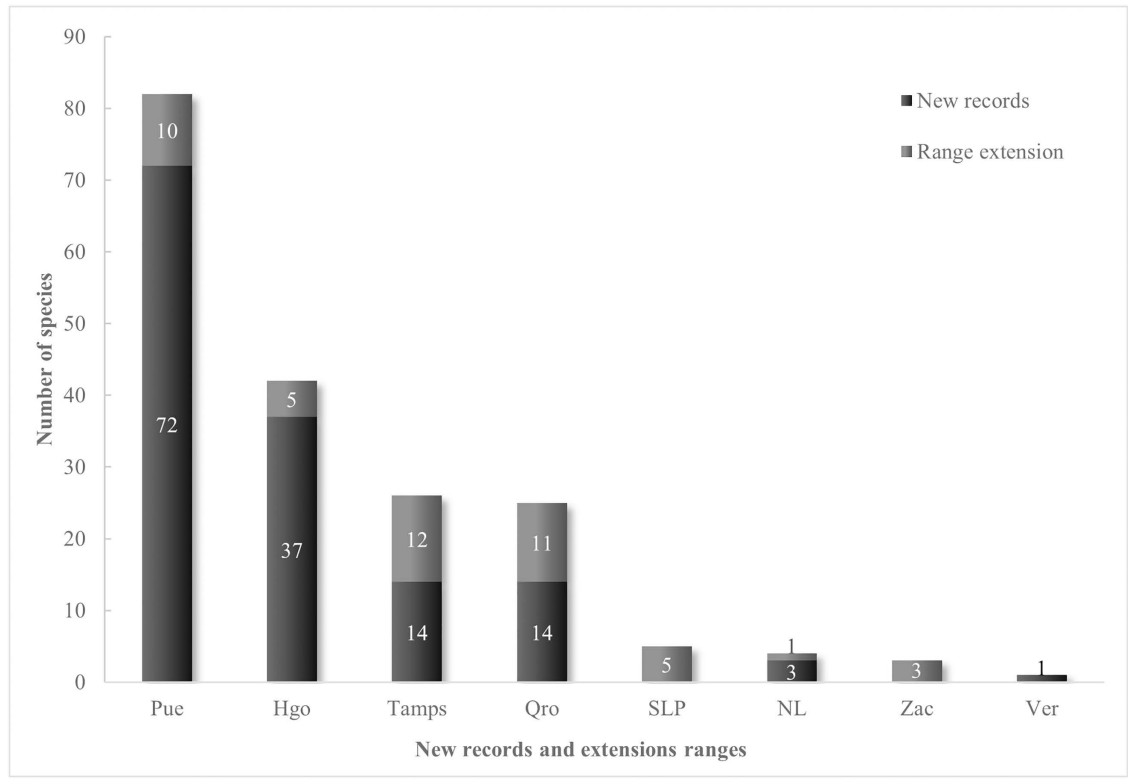

**Fig 3. New pteridophyte records for the Sierra Madre Oriental (123 species) and species with an extended distribution in other states from the SMOR (40).**

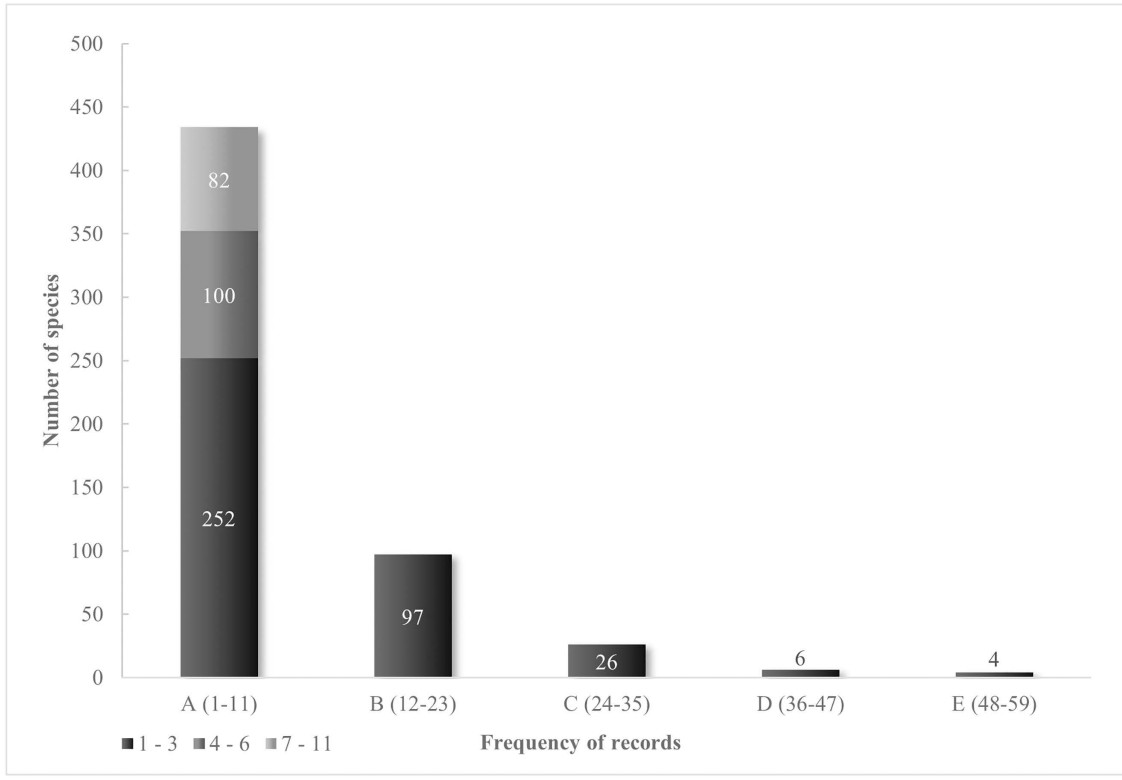

**Fig 4. Record frequency for each species represented in the Sierra Madre Oriental.**

## Grid cell species richness and endemism indices

We analyzed 86 grid cells from the SMOR (Fig 5A), each containing at least one record. Species richness values ranged from 1 to 307 species per grid cell (Fig 5B). The main center of pteridophyte diversity in the southern part of the SMOR (states of Puebla, Hidalgo, Querétaro, and Veracruz) consisted of nine neighbouring grid cells with the highest species richness scores (>90), ranging from 96 to 307 species (Fig 5B). The tenth grid cell with the highest richness score (95) is located in Nuevo León (BB). Among the top-scoring grid cells, BX, CC, CD, and CG have the highest number of species (>200). These high-richness grid cells contrast with other parts of the SMOR, where collections are sparse or absent, as in 13 grid cells with only one fern or lycophyte species and eight grid cells with just two taxa recorded (Fig 5B).

*Polypodium plesiosorum* (27), *Adiantum capillus-veneris* and *Astrolepis sinuata* (26 each), *Pleopeltis polylepis* (25), and *Astrolepis integerrima* (23) were found in more grid cells, followed by *Pelazoneuron ovatum* and *Pleopeltis guttata* (21) (Table 2). Conversely, 150 species are only represented in one grid cell. Ferns and lycophytes are mainly distributed in temperate forests, such as oak and pine-oak forests, and in tropical montane cloud forests. However, some species prefer other types of Mexican vegetation that receive less rainfall, such as the arid scrub and tropical deciduous forest. Temperate landscapes mainly influence the key distribution patterns observed in this study.

We undertook an effort-weighted sampling method for our manuscript. In our study using the Chao1 richness estimator, we identified 567 species. According to the estimator, the average estimated richness at this sampling level was about 670 species, with a 95% confidence interval of approximately 632–729 species (Fig 6). The difference between the lower and upper confidence limits is roughly 100 species, indicating moderate uncertainty around the estimate, as expected in communities with many rare or low-detectability taxa.

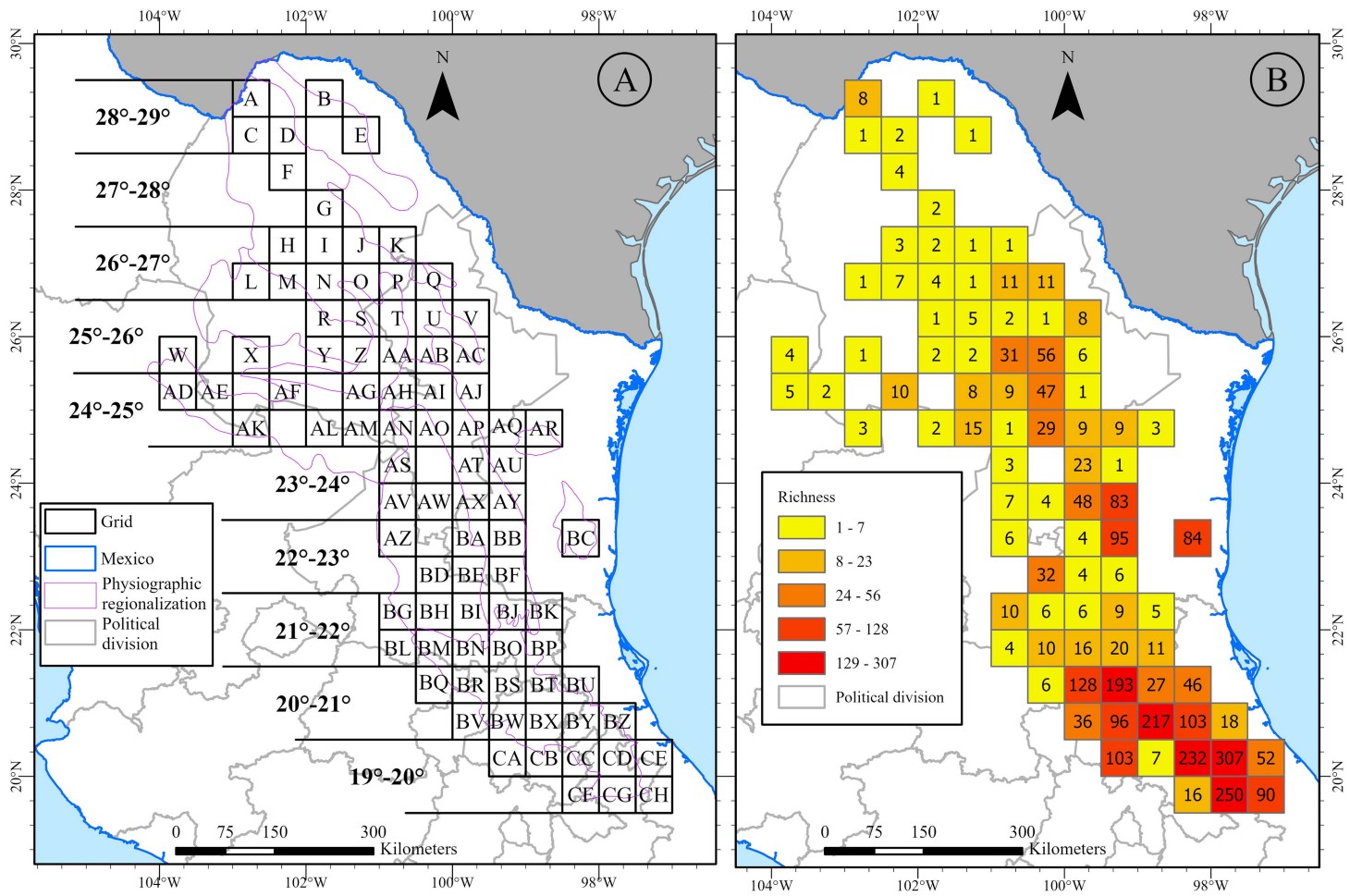

**Fig 5. A. Division of the Sierra Madre Oriental into grid cells of 30 minutes by side and latitudinal belts of one geographical degree. B.** Species richness of the pteridoflora in the SMOR. Map created by the authors using ArcGIS Pro. Physiographic regionalization (SMOR) and Political Division layers were sourced from INEGI (https://www.inegi.org.mx/), used under the Terms of Free Use of Information which permit commercial utilization.

Comparing the observed richness (567 species) to the Chao1 mean estimate (≈670 species), our data falls short by about 100 species from the estimated total richness. This suggests that our sampling completeness is roughly 85%, which is relatively high, because Chao1 uses the number of very infrequently observed species to infer unseen diversity, tending to produce higher richness estimates when rare species are relatively common in the sample [89,90]. These curves show that we've captured more than 85% of the expected species at a sampling site, indicating that this type of analysis is feasible [91].

Overall, these results suggest that although the sampling did not fully capture the total estimated richness, it still documented a significant portion of the community. Using Chao1, the accumulation curves for fern and lycophyte species in the SMOR did not reach a defined asymptote, remaining above the observed richness.

## Endemism indexes

Figs 7A and 7B display the values of weighted endemism (WE) and corrected weighted endemism (CWE) for ferns and lycophytes of the SMOR. For WE, values ranged from 0.04 to 115.8 across the grid cells analyzed (Fig 7A). The WE

**Table 2. Species with more records and recorded in more grid cells of the Sierra Madre Oriental.**

| Species | Number of records | Grid cells |
|---|---|---|
| *Pleopeltis polylepis* (Roemer ex Kunze) T. Moore | 57 | 25 |
| *Pleopeltis plebeia* (Schltdl. & Cham.) A. R. Sm. & Tejero | 56 | 20 |
| *Polypodium plesiosorum* Kunze | 53 | 27 |
| *Phlebodium pseudoaureum* (Cav.) Lellinger | 52 | 17 |
| *Tectaria heracleifolia* (Willd.) Underw. | 47 | 20 |
| *Campyloneurum angustifolium* (Sw.) Fée | 41 | 15 |
| *Adiantum capillus-veneris* L. | 40 | 26 |
| *Pelazoneuron ovatum* (R.P. St. John) A.R.Sm. & S.E. Fawc. | 40 | 21 |
| *Llavea cordifolia* Lag. | 36 | 19 |
| *Pleopeltis crassinervata* (Fée) T. Moore | 36 | 10 |
| *Adiantum tenerum* Sw. | 35 | 13 |
| *Asplenium monanthes* L. | 35 | 18 |
| *Astrolepis sinuata* (Lag. ex Sw.) D.M. Benham & Windham | 35 | 26 |
| *Blechnum appendiculatum* Willd. | 34 | 15 |
| *Adiantum poiretii* Wikstr. | 33 | 15 |
| *Pleopeltis acicularis* (Weath.) A.R. Sm. & T. Krömer | 33 | 11 |
| *Pteridium aquilinum* (L.) Kuhn | 33 | 13 |
| *Anemia mexicana* Klotzsch | 31 | 20 |
| *Notholaena candida* (M. Martens & Galeotti) Hook. | 31 | 20 |
| *Pteris cretica* L. | 31 | 19 |
| *Astrolepis integerrima* (Hook.) D.M. Benham & Windham | 30 | 23 |
| *Pleopeltis polypodioides* (L.) E.G. Andrews & Windham | 30 | 20 |
| *Anemia adiantifolia* (L.) Sw. | 29 | 19 |
| *Pleopeltis guttata* (Maxon) E. G. Andrews & Windham | 29 | 21 |
| *Polypodium arcanum* Maxon | 29 | 15 |
| Remaining taxa | 3,543 | 2,289 |
| **Total** | **4,479** | **2,757** |

values showed a pattern similar to that of species richness. We found that nearly all grid cells with the highest species richness scores also recorded the highest WE values (>16), except for the grid cell BW.

The corrected weighted endemism values ranged from 0.05 to 0.48 (Fig 7B), excluding grid cells with only one species, as previously established. The grid cells with the highest CWE values (>0.25) are spread throughout the SMOR, and only one grid cell (CD) with the highest species richness is included.

## Elevation patterns

The pteridoflora in the SMOR is found across the entire elevation range of this mountain range (100–3700 m). As is typical in Neotropical mountains, species richness in the SMOR exhibits a hump-shaped elevational pattern, with the highest diversity at mid-elevations (Fig 8). The greatest number of species occurs between 1000 and 1500 m (455 species), decreasing at both higher and lower elevations.

## Latitudinal patterns

We believe that the pteridoflora of the SMOR did not strictly follow a latitudinal pattern. Species diversity does not consistently decrease or increase with latitude (Fig 9). Our analysis of latitude within 1 ° geographic belts (Fig 5A) revealed

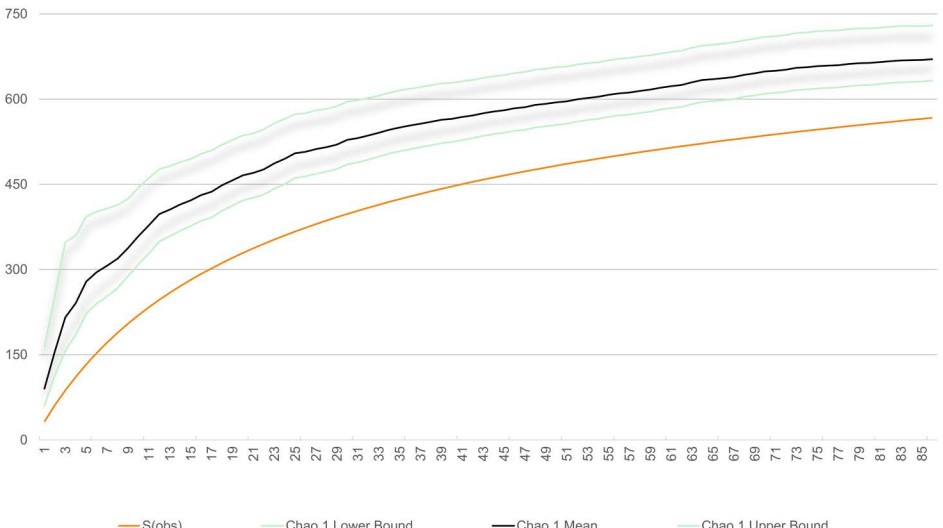

**Fig 6. Species accumulation curve for ferns and lycophytes of the Sierra Madre Oriental (SMOR) using the Chao1 richness estimator.** The observed species richness is shown as a solid line. The Chao1 estimator is represented by three curves: the mean estimated richness and its lower and upper confidence limits (95%).

low richness at the northern extreme (28°-29° N, 11 species) and high richness at the southern extreme (19°-20° N, 469 species). Another peak in richness occurs at the 24 °- 25 ° N belt (with 350 species). The decline in species richness is more apparent from the 24°-25° N belt to the northernmost point (28°-29° N).

### Geographic distribution patterns

Regarding the geographic distribution of ferns and lycophytes inhabiting the SMOR, the Neotropical and Mesoamerican components are the best represented, with 238 and 119 species, respectively (Fig 10) (see S1 Appendix). They are followed by the endemic component to Mexico, which has 64 species. Other significant components include the Nearctic and American, represented by 44 and 41 species. The high proportion of Neotropical and Mesoamerican components in the SMOR enhances the significance of this mountain range within the Mexican Transition Zone. Each remaining geographic component has fewer than 30 species (Fig 10).

We recorded 21 species that are endemic to the SMOR: *Asplenium dianae*, *Asplenium semipinnatum*, *Asplenium ultimum*, *Bryodesma carnerosanum*, *Gaga apiacea*, *Gaga hintoniorum*, *Goniopteris schaffneri*, *Isoetes tamaulipana*, *Myriopteris chipinquensis*, *Myriopteris maxoniana*, *Notholaena brachycaulis*, *Notholaena brevistipes*, *Notholaena jacalensis*, *Notholaena leonina*, *Pellaea notabilis*, *Pellaea ribae*, *Pleopeltis ×gracilis*, *Pleopeltis ×pueblensis*, *Pleopeltis fallacissima*, *Pulviniella gypsophila*, and *Woodwardia martinezii*. At least five other species are closely associated with the SMOR and have geographic distributions almost restricted to this mountain chain (S1 Appendix).

### Risk categories

There are 20 fern species and two lycophytes listed in the Mexican Official Norm NOM-059 [85]. These species are shown in Table 3. Among them, 12 are classified as special protection category (Pr), seven as threatened (A), and three as at risk of extinction (P). The three species are *Cyathea costaricensis, Cyathea godmanii*, and *Selaginella porphyrospora*. According to the Red Lists of Threatened Species by the IUCN [86], seven fern taxa are categorized as least concern (LC),

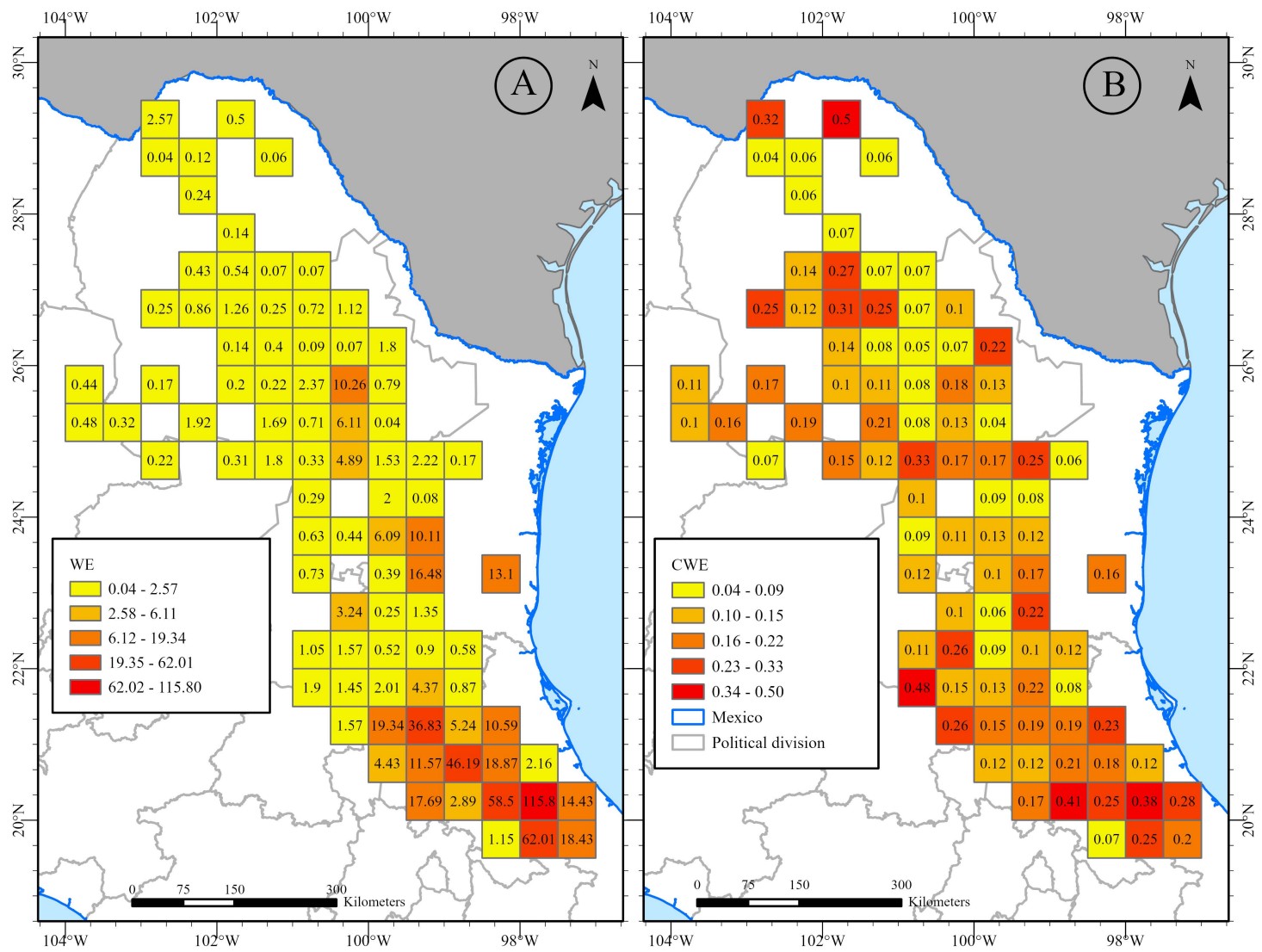

**Fig 7. A. Weighted endemism (WE). B.** Corrected weighted endemism (CWE) in the Sierra Madre Oriental divided in grid cells of 30 minutes. Map created by the authors using ArcGIS Pro. Political Division layers were sourced from INEGI (https://www.inegi.org.mx/), used under the Terms of Free Use of Information which permit commercial utilization.

indicating a low risk of extinction (see Table 3). Only members of Cyatheales are included in Appendix II of CITES [87], comprising eight Cyatheaceae and two Dicksoniaceae (see Table 3).

## Discussion

Our study increased the number of ferns and lycophytes recorded in the SMOR compared to earlier floristic surveys conducted in this Mexican mountain range [18,19]. We documented 567 species, including 511 ferns and 56 lycophytes, thereby expanding our understanding of these plant groups and the Mexican pteridoflora. Our findings reflect an increase of 211 species compared to the 356 species in the SMOR, as documented by [12,19], which reported a total of 1,055 species in Mexico. Tejero's personal database (2025) includes 1,124 taxa, encompassing subspecies, varieties, and hybrids.

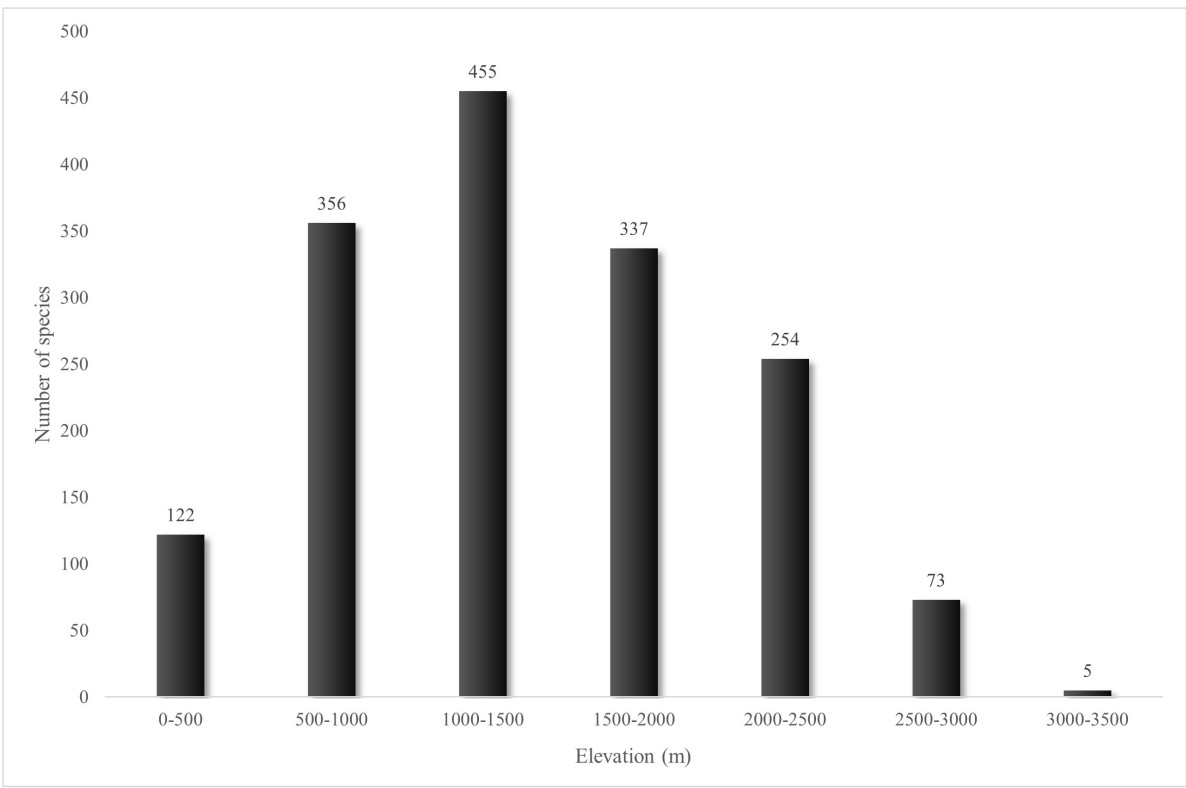

**Fig 8. Elevational distribution of the pteridophyte species richness in the Sierra Madre Oriental.**

Given this diversity, the new total of species recorded here in the SMOR represents approximately 50–54% of the Mexican pteridoflora. This study also documents 60 names and/or species that were excluded following taxonomic refinement and review.

Our study, mainly based on specimen revisions from herbaria and fieldwork, uncovers new records and range extensions for over 100 species of pteridoflora in the SMOR. Our revision identifies 123 species in 62 genera as new for the area and 40 species in 25 genera showing range extensions within this mountain chain. *Elaphoglossum* and *Hymenophyllum* are the fern genera with the most new records in the SMOR; a similar situation occurs in an inventory of the pteridoflora in Veracruz [43], where *Elaphoglossum* and filmy ferns were also reported with the highest number of new records among ferns. Our findings confirm that ongoing fieldwork in the mountains of the Mexican Transition Zone could uncover new records and range extensions of the Mexican pteridoflora [43], even in regions that have already been floristically studied, such as the SMOR (e.g., [18,19]).

The geographic position, its alignment with the oceanic edge, and the size of the SMOR influence the observed patterns of differential richness. A common pattern of species richness along the latitudinal gradient, which shows an increase from the north to the tropics [3,41,92], was disrupted. Despite over 200 years of research, the factors driving geographic patterns of species richness remain largely misunderstood [93]. In the case of the SMOR, an intermediate zone of the gradient (24°–25° N) was found to have a relatively high proportion of pteridoflora compared to the southern latitudinal zones (19°–20° N), with 469 species (especially Puebla; 31.9 species/km²) and the northern zone with the lowest (Coahuila 2.7 species/km²).

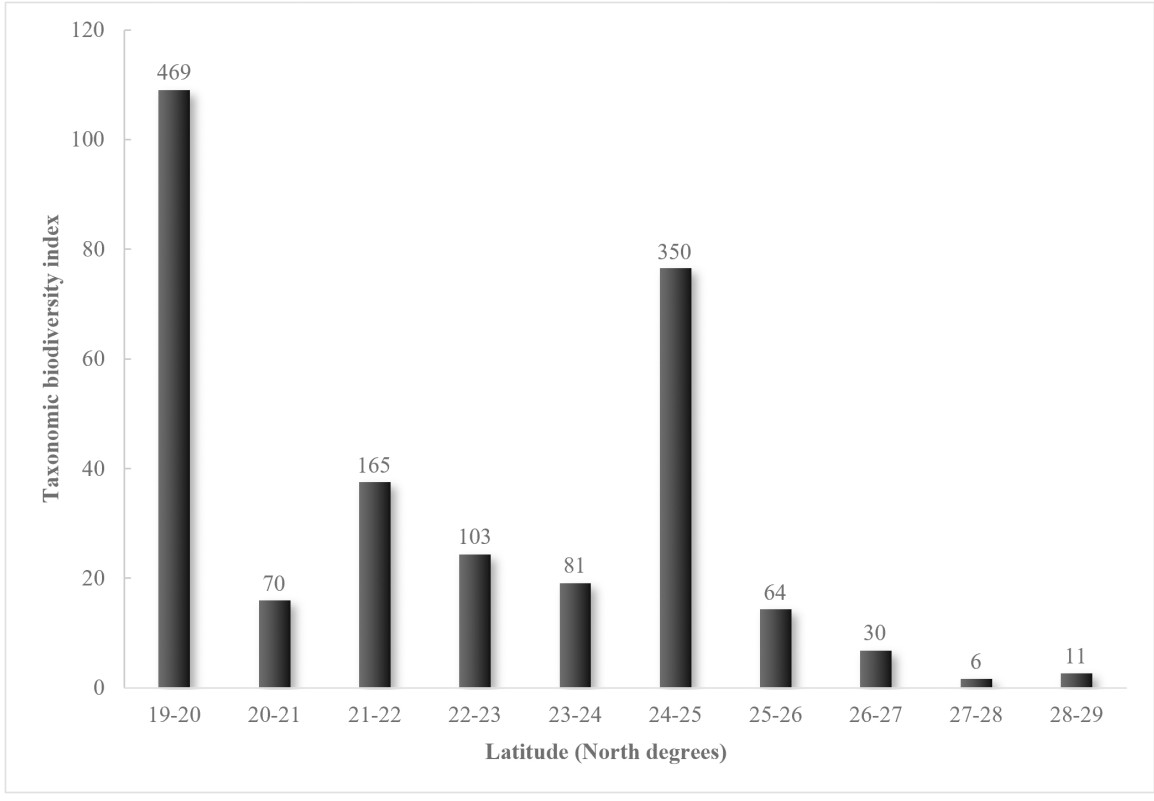

**Fig 9. Latitudinal belts of one geographic degree and their values of the taxonomic biodiversity index in the Sierra Madre Oriental.** The numbers over each column represent the species richness recorded in each latitudinal belt.

Most likely, the ecological factor responsible for this intermediate peak in species richness can be attributed to the combination of the broader SMOR band with the effect of orographic shadow, which creates an unparalleled environmental heterogeneity in the mountain range. Additionally, the states on the oceanic slope are more diverse (Querétaro 17.6 species/km²) than those on the interior (Guanajuato 1.2 species/km²). By analyzing both the latitude and orographic shadow patterns, we can explain the high richness at southern latitudes (20–21°; Hidalgo, Puebla and Veracruz) and the intermediate richness at latitudes 25–26°, where richness increases from west to east across the mountain (Durango, Coahuila and Zacatecas; and Nuevo León, Tamaulipas) (Fig 9).

As observed in the mountainous regions of southeastern Mexico [6], the SMOR shows a strong Neotropical influence. Many Neotropical and Mesoamerican elements are well-represented in the Sierra Madre del Sur and the Chiapas Highlands in southeastern Mexico. In the SMOR, the flowering plants display a marked Nearctic influence [18], contrasting with the predominant Neotropical affinities of the pteridoflora reported here. Compared to the Sierra Madre Occidental [14], the SMOR showcases the typical association of Neotropical ferns in montane ecosystems due to its high humidity levels, like those found on the Gulf of Mexico slope, and hosts several of the most diverse epiphytic fern clades, including *Asplenium*, *Elaphoglossum*, Hymenophyllaceae, and Polypodiaceae [3]. The diversity of pteridoflora in the SMOR and the presence of endemic species support the idea of a tropical Mexican fern hotspot, with the SMOR forming part of one of the main regional hotspots in the Neotropics recognized by [92].

Pteridaceae, Dryopteridaceae, and Polypodiaceae are regarded as some of the most diverse fern families worldwide. Some genera and subfamilies reflect the dominant ecological zones in the SMOR. Pteridaceae is a successful group

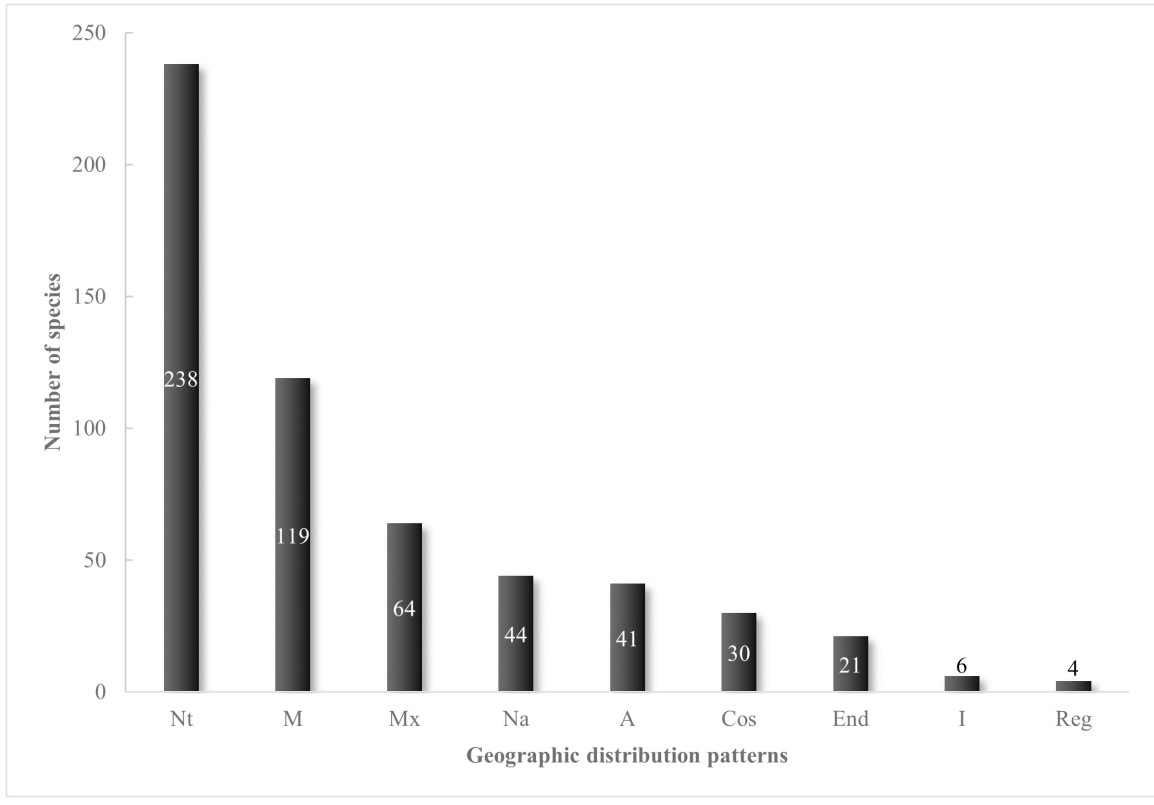

**Fig 10. Geographic distribution patterns of the pteridoflora in the Sierra Madre Oriental.** The abbreviations are: (Cos) Cosmopolitan; **(A)** American; (Na) Nearctic; (Nt) Neotropical; **(M)** Mesoamerican; (Mx) Endemic to Mexico; (End) Endemic to the SMOR; (Reg) Regional endemic; **(I)** Introduced.

represented there by 131 species and is classified into three major bioclimatic groups: adiantoids, pteroids, and cheilanthoids. The cheilanthoid ferns (*Argyrochosma, Astrolepis, Gaga, Myriopteris, Notholaena, Pellaea*) are primarily found on the north and western slopes facing the SMOR, which features xerophytic and seasonal temperate vegetation. Conversely, the adiantoid ferns with *Adiantum* as their most representative genus, are mainly present in the eastern part of the SMOR, characterized by a warm, humid climate. Lastly, the remnant pteroid group (*Pteris*, *Vittaria*) is predominantly located in the temperate, humid montane region [94].

The other two fern families with high species richness in the SMOR are Dryopteridaceae and Polypodiaceae, mainly inhabiting the humid montane forests on oceanic slopes [19,95]. In the SMOR, Polypodiaceae includes 18 genera and 76 species, exhibiting the greatest variation in form among ferns, including epiphytic species [95]. Some genera, such as *Campyloneurum, Pecluma* (both with 10 species), *Polypodium* (11 species), and *Pleopeltis* (24 taxa), are mainly represented by epiphytic forms in the SMOR [95]. Dryopteridaceae are primarily found in temperate and tropical montane habitats [96]; they comprise 12 genera and 68 species, mostly terrestrial, except for *Elaphoglossum*, a diverse genus with 30 species [97]. Zotz [98] indicates that approximately 66% of all *Elaphoglossum* species are epiphytes according to its preference. The colonization of the vertical space of trees by *Elaphoglossum*, *Pleopeltis*, *Polypodium*, and members of the Hymenophyllaceae family significantly contributes to the high diversity of these forests compared to other regions of the warm zone. This pronounced richness of epiphytic species in humid montane forests is a consistent pattern observed in other Mexican regions, including central Veracruz, Los Tuxtlas, and the Sierra de Juárez in Oaxaca. Furthermore, this elevated epiphyte diversity is considered a primary driver of the unimodal elevational pattern observed in fern species

**Table 3. Threatened species of ferns and lycophytes of the Sierra Madre Oriental and their risk categories.**

| Family | Species | NOM-059 | IUCN | CITES |
|---|---|---|---|---|
| Aspleniaceae | *Asplenium auritum* Sw. | A | | |
| | *Asplenium formosum* Willd. | | LC | |
| | *Asplenium serratum* L. | A | | |
| Culcitaceae | *Culcita coniifolia* (Hook.) Maxon | Pr | | |
| Cyatheaceae | *Alsophila firma* (Baker) D. S. Conant | Pr | | Appendix II |
| | *Cyathea bicrenata* Liebm. | Pr | | Appendix II |
| | *Cyathea costaricensis* (Mett. ex Kuhn) Domin | P | | Appendix II |
| | *Cyathea fulva* (M. Martens & Galeotti) Fée | Pr | | Appendix II |
| | *Cyathea godmanii* (Hook.) Domin | P | | |
| | *Cyathea microdonta* (Desv.) Domin | Pr | | Appendix II |
| | *Cyathea myosuroides* (Liebm.) Domin | Pr | | Appendix II |
| | *Cyathea schiedeana* (C. Presl) Domin | Pr | | Appendix II |
| | *Cyathea tuerckheimii* R. M. Tryon | Pr | | |
| | *Sphaeropteris horrida* (Liebm.) R. M. Tryon | Pr | | Appendix II |
| Dicksoniaceae | *Dicksonia navarrensis* Christ | Pr | | Appendix II |
| | *Lophosoria quadripinnata* (J. F. Gmel.) C. Chr. | | | Appendix II |
| Equisetaceae | *Equisetum hyemale* L. var. *affine* (Engelm.) A. A. Eaton | | LC | |
| Lycopodiaceae | *Phlegmariurus dichotomus* (Jacq.) W.H. Wagner | A | | |
| Marattiaceae | *Marattia laxa* Kunze | Pr | | |
| | *Marattia weinmanniifolia* Liebm. | Pr | | |
| Osmundaceae | *Osmunda spectabilis* Willd. | | LC | |
| Polypodiaceae | *Campyloneurum phyllitidis* (L.) C. Presl | A | | |
| | *Serpocaulon triseriale* (Sw.) A. R. Sm. | A | | |
| Psilotaceae | *Psilotum ×complanatum* Sw. | A | | |
| | *Psilotum flaccidum* Wall. | A | | |
| Pteridaceae | *Adiantum capillus-veneris* L. | | LC | |
| | *Pteris vittata* L. | | LC | |
| Selaginellaceae | *Selaginella porphyrospora* A. Braun | P | | |
| Thelypteridaceae | *Cyclosorus interruptus* (Willd.) H. Itô | | LC | |
| | *Goniopteris tetragona* (Sw.) C. Presl | | LC | |

Abbreviations. Risk categories of the NOM-059: (A) Threatened; (Pr) Special protection; (P) Endangered. IUCN: (LC) Least concern category.

richness. This pattern typically peaks at mid-elevations, where optimal moisture and microclimatic stability provide ideal conditions for the proliferation of these lineages [41,99–101]. Among lycophytes, the richest family in the SMOR is Selaginellaceae, represented solely by *Selaginella*, with 41 species [12]. The list of lycophytes presented in this study is based on [65]. Members of Selaginellaceae thrive under a wide range of climates and soil types. Species with homophyllous leaves (*Bryodesma*) and heterophyllous, rosette-forming leaves (*Lepidoselaginella*) are poikilohydric, drought-adapted xerophytes [102,103]. The heterophyllous species are diverse in tropical forests worldwide, predominantly in secondary environments [104]. Both mesic and tropical-xeric ecological zones are well represented in the SMOR. The same is true for other Mexican montane ranges, such as the Sierra Madre Occidental [14] and the Sierra Madre del Sur [6], which, along with the Trans-Mexican Volcanic Belt and SMOR, form the Mexican Transition Zone sensu [105]. Conversely, in the southern Mexican mountains with homogeneous mesic environments, such as the Chiapas Highlands, there are only 24 species [22].

In the heatmap, the concentration of collection records is located in the central part of the country, corresponding to the southern part of the SMOR. This map shows that collecting effort is not randomly distributed, as is often the case in biodiversity studies [106], and the disproportion of records in the southern part of the SMOR may be linked to accessible geographic locations such as roadsides and urban areas, as well as proximity to research institutions in central Mexico with active botanical research. This pattern has been previously suggested for biodiversity studies based on herbarium specimens [106–108]. The higher number of collection records and species of ferns and lycophytes in the southern SMOR can also be explained by the higher rainfall and warmer temperatures in this region [28]. These climatic conditions support the presence and abundance of these vascular plants in the tropical montane cloud forests, which are preferred by many botanists for the study and field collection of Mexican ferns and lycophytes, as in other parts of the country [5,95].

The grid cells with high species richness scores include families and genera that typically inhabit humid environments [4]. The convergence of certain families, such as Lycopodiaceae, Selaginellaceae, Aspleniaceae, Athyriaceae, Cyatheaceae, Dryopteridaceae, Hymenophyllaceae, Polypodiaceae, Pteridaceae, and Thelypteridaceae in these highly diverse grid cells reflects complex plant assemblages in these areas, due to the presence of different phylogenetic lineages of the Mexican pteridoflora, linked to epiphytic and terrestrial lifestyles [93].

The estimator values used showed similar behavior and remained close to the observed values, indicating that a good sampling of the pteridophyte flora was achieved [90]. Although the number of pteridophyte species reported in this study is much higher than that recorded in previous research (e.g., [19]), the analysis of the species accumulation curves suggests that more species can still be recorded to complete the list of ferns and lycophytes of the SMOR.

The WE values obtained for ferns and lycophytes in the SMOR are very similar to the species richness. Nearly all grid cells with high species richness scores (except BW) also showed high WE index values. This correlation between species richness and WE values has been suggested in previous studies [27,73,74]. The CWE index emphasizes species with limited distribution, so only one grid cell (CD) is represented in the WE with the highest values. The grid cells with high CWE values (≥0.25 and containing two or more recorded species) are spread across different zones of the SMOR and are influenced by the geographic distribution of the species present in these grid cells, some of which are endemic to the SMOR, such as *Asplenium dianae, A. semipinnatum, A. ultimum, Notholaena brachycaulis,* and *Pellaea ribae.*

We regard the CWE index as an essential measure of endemism because it is independent of species richness [69,73,74], as it includes species endemic to the SMOR and Mexico. We recommend that grid cell CD—located primarily in the Sierra Norte de Puebla and extending into a small portion of Veracruz—should be a priority area for conserving the Mexican pteridoflora due to its unparalleled diversity (307 species, supported by both WE and CWE indexes). Although this area already overlaps with the established Natural Protected Areas (such as the Carmen Serdán National Park and the Río Necaxa and Kowtahyolo Natural Resource Protection Areas), its exceptional richness highlights the urgent need to maintain connectivity and expand current conservation strategies across the entire grid cell. Further research and fieldwork are needed in grid cells with the highest CWE scores to boost the number of records and species in these areas.

Following the "World Ferns" sensu [12], some species are found in only one Mexican state. Our study revealed that many of these species (e.g., *Pecluma liebmannii* and *Notholaena brevistipes*), previously recorded as endemic to just one state, are also found in another Mexican state. These two species, along with *Ctenitis bullata, Pleopeltis ×melanoneuron,* and *Selaginella subrugosa*, are documented in our study for the first time in one to three Mexican states, changing their endemism status from local endemic to regional endemic.

Considering the high diversity of the pteridoflora in the SMOR (567 species), we observed that only a small percentage of these vascular plants are classified as threatened by international and national agencies. Regrettably, only 22 species are listed under some risk category in the NOM-059 [85] by the Mexican government, and seven are in the least concern (LC) category of the IUCN [86]. This low number of threatened species suggests that other fern and lycophyte taxa need to be assessed according to the criteria of the IUCN [86] and the NOM-059 [85], such as regional endemics and those

species limited to the tropical montane cloud forests [109], to determine their potential inclusion in the National and International Red Lists.

We suggest that the 19 endemic species should be assessed for inclusion in NOM-059, as they are regionally endemic and have few documented instances. None of these species is listed on the Red List, and their traits exceed those considered for some species already included. Candidate species include taxa from Pteridaceae (genus *Notholaena*, with five species) from semiarid zones with xerophytic vegetation in northern SMOR.

The *Phlegmariurus* species (Lycopodiaceae) are notable and are not necessarily restricted to SMOR. [110] proposed that all Mexican *Phlegmariurus* species be regarded as threatened or endangered. These species inhabit undisturbed areas of mature pine-oak and humid montane forests, often represented by only a few individuals at each site [111]. The NOM-059 includes only one species (*P. dichotomus*), categorized as threatened. The genus *Phlegmariurus* in Mexico comprises 13 species, seven of which are found in SMOR. These lycophytes are rare or absent in cleared areas because they are epiphytes of mature host canopy trees; they are located in limited localities and collections [106], such as *P. linifolius* (5 records), *P. pringlei* (5), *P. pithyoides* (3), and *P. orizabae*, documented from only a single specimen collected in Puebla.

The family Cyatheaceae is a threatened component in the SMOR. The mountainous regions of the Neotropics host most of the species richness of American tree ferns [71]. These vascular plants are critically endangered, facing threats from deforestation, exploitation, and global climate change [71,112,113]. Most are listed in the NOM-059, but their risk categories need reassessment and updating [71,114]. Unfortunately, deforestation and land-use change in the SMOR have reduced tropical mountain forest areas due to agriculture and livestock raising [17,114], threatening the survival of these ferns. The extraction of tree fern stems for handicrafts and as growing medium or substratum for orchids and other plants poses a serious threat to these ferns [112,113], and some conservation initiatives have been proposed within the SMOR, e.g., to sell young tree ferns for horticultural use because such plants are abundant in disturbed areas and their transplantation from roadsides to safe sites under shade conditions is suggested [115]. According to Appendix II of CITES, tree ferns are not necessarily at risk of extinction but could become threatened if their trade is not properly regulated. Therefore, the Mexican government must establish trade regulations for Cyatheaceae species to prevent their exploitation and use that threaten their survival.

## Conclusions

In the SMOR, [18] recorded 8,472 vascular plant species, suggesting that this mountain range has greater floristic richness than any other mountainous region in Mexico. Conversely, [19] documented 6,981 vascular plants in the SMOR, including 356 species of ferns and lycophytes; [18] identified at least 104 as 'characteristic' species of the SMOR without listing the total number of taxa. Our findings include 211 additional species of pteridophytes, exceeding those reported by [19], and thus increasing the overall number of vascular plants in the SMOR.

Herbarium data are crucial for understanding plant biogeography; however, sampling biases and coverage gaps are common in biogeographic studies [108]. Our analysis, mainly based on herbarium specimens, is therefore influenced by these biases. Some areas within the SMOR boundaries lack records of ferns or lycophytes, and 13 grid cells contain only one species. These low-diversity areas may be related to unexplored zones and insufficient sampling, as observed in many sectors of Mexico [67,69] and the Neotropics [1,111]. Widespread species are more frequently collected than rare or microendemic species. Future collection efforts in poorly sampled areas of the SMOR are essential to enhance the fern and lycophyte inventory, emphasising the importance and ongoing need for fieldwork [1].

This study provides a detailed inventory of ferns and lycophytes from one of the world's most biodiverse regions, recognized as part of a biodiversity hotspot [92]. This mountain range serves as a transition zone between different biogeographic regions (Neotropical and Nearctic), with diverse climatic, edaphic, and geological conditions and variations in latitude and elevation. This results in high environmental heterogeneity, leading to significant plant diversity [18]. Based on

taxonomic dominance, the most prominent environments are the subhumid temperate (*Pleopeltis* zone) in the middle and high mountains, followed by the semiarid (*Notholaena* zone) on the northern and western slopes, the warm humid (*Adiantum* zone) on the eastern slope, and humid temperate (*Hymenophyllum* zone) in the southern part of the SMOR.

Our findings show that the greatest diversity of ferns and lycophytes is found in the southwestern part of the SMOR, corresponding primarily in the Sierra Norte de Puebla and extending into a small portion of Veracruz. This is confirmed by climate data: the southern part of the SMOR receives the most rainfall (>1,200 mm, up to 4,000 mm in Cuetzalan, Puebla) and has the warmest temperatures (between 22 and 26 °C) [28]. Both climate conditions support the presence and growth of many fern and lycophyte species.

Another distinct region is the central part of the SMOR in the state of Tamaulipas (grid cells AY, BB, and BC), which includes the El Cielo Biosphere Reserve and the Sierra de San Carlos. The climate in this section of the SMOR, covering Tamaulipas, features temperate conditions (12–18 °C), with rainfall ranging from 600 to 1,200 mm [28], supporting species with Nearctic affinities. Due to the unique biology of lycophytes and ferns, involving two different phases in their life cycle-one sensitive to humidity [4–6] and their stomata's low reactivity [7–9], it is a taxocenosis that functions as an indicator of local environmental conditions. Appropriate vegetation cover and moisture enhance the richness and abundance of pteridoflora. As a result, the distribution patterns observed in this study can serve as a foundation for predicting future environmental impacts, whether from deforestation or climate change.

## Supporting information

**S1 Appendix. Checklist of ferns and lycophytes of the Sierra Madre Oriental and excluded taxa.** This file contains two sheets: 1) A comprehensive checklist detailing species presence by state, subprovince, geographic distribution, and elevation range. 2) A list of excluded names and synonyms for taxa that were not verified or do not naturally occur in the study area.
(XLSX)

## Acknowledgments

We are grateful to the curators and staff of the herbaria mentioned in the text for their courtesy during our review of specimens. The HUAP, through Amparo Cerón, helped develop a database from northern Puebla. The IEB, through Brenda Bedolla, helped develop the database from northeastern Querétaro related to the SMOR. CONABIO, through Susana Ocegueda, helped develop the database from Veracruz. Jonathan D. Amith, co-author, contributed to this research effort as director of a long-term floristic and ethnobotanical project in the Sierra Nororiental de Puebla, during which > 600 ferns and allies were collected, principally in the municipalities of Cuetzalan del Progreso (182), Zacapoaxtla (157) Zautla (72), Zoquiapan (58), and Zongozotla (41). Fieldwork teams were led by Canek Ledesma Corral and Miriam Jiménez Chimil and included Ceferino Salgado Castañeda, Mariano Gorostiza Salazar, Eleuterio Gorostiza Salazar, and Osbel López Francisco.

## Author contributions

**Conceptualization:** J. Daniel Tejero-Díez, Perla V. Rodríguez-Sánchez, Isolda Luna-Vega.

**Data curation:** J. Daniel Tejero-Díez, Perla V. Rodríguez-Sánchez, Leccinum J. García-Morales.

**Formal analysis:** Raúl Contreras-Medina, Julio Cesar Ramírez-Martínez, Isolda Luna-Vega.

**Funding acquisition:** Isolda Luna-Vega.

**Investigation:** Raúl Contreras-Medina, Julio Cesar Ramírez-Martínez, Isolda Luna-Vega.

**Methodology:** Raúl Contreras-Medina, Julio Cesar Ramírez-Martínez, Isolda Luna-Vega.

**Project administration:** Isolda Luna-Vega.

**Resources:** Isolda Luna-Vega.

**Software:** Julio Cesar Ramírez-Martínez.

**Supervision:** Jonathan D. Amith, Isolda Luna-Vega.

**Validation:** Jonathan D. Amith.

**Visualization:** Julio Cesar Ramírez-Martínez.

**Writing – original draft:** J. Daniel Tejero-Díez, Perla V. Rodríguez-Sánchez, Raúl Contreras-Medina, Isolda Luna-Vega.

**Writing – review & editing:** J. Daniel Tejero-Díez, Raúl Contreras-Medina, Julio Cesar Ramírez-Martínez, Celia Sanginés-Franco, Isolda Luna-Vega.

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
