## [Decision Letter · Decision Letter 0]

29 Oct 2025

PONE-D-25-44575Richness and biogeography of Pteridoflora in montane forests of eastern MexicoPLOS ONE

Dear Dr. Luna-Vega,

Thank you for submitting your manuscript to PLOS ONE. After careful consideration, we feel that it has merit but does not fully meet PLOS ONE’s publication criteria as it currently stands. Therefore, we invite you to submit a revised version of the manuscript that addresses the points raised during the review process.

We look forward to receiving your revised manuscript.

Kind regards,

Marcela Pagano, Ph.D, M.D.

Academic Editor

PLOS ONE

Journal Requirements:

“We are grateful to the curators and staff of the herbaria mentioned in the text for their courtesy during our review of specimens. Project DGAPA-PAPIIT IN219424 funded the project.”

“Project DGAPA-PAPIIT IN219424 funded the project.”

“Project DGAPA-PAPIIT IN219424 funded the project.”

Reviewers' comments:

Reviewer's Responses to Questions

**Comments to the Author**

1. Is the manuscript technically sound, and do the data support the conclusions?

Reviewer #1: Yes

Reviewer #2: Yes

2. Has the statistical analysis been performed appropriately and rigorously? 

Reviewer #1: Yes

Reviewer #2: I Don't Know

3. Have the authors made all data underlying the findings in their manuscript fully available?

Reviewer #1: No

Reviewer #2: Yes

4. Is the manuscript presented in an intelligible fashion and written in standard English?

Reviewer #1: Yes

Reviewer #2: Yes

5. Review Comments to the Author

Reviewer #1: I have reviewed your manuscript titled "Richness and biogeography of Pteridoflora in montane forests of eastern Mexico" and found it to be well-written. The methods are appropriate, and the results and discussion are well presented. I have only two concerns:

1. You need to explicitly explain how you obtained the geographical coordinates for the specimens that lack them.

2. The dataset used for all the analyses (species with their geographical coordinates) is not available. I encourage you to make this information public.

Reviewer #2: Methodologically, I think the article is well presented.

Although it is presented from a descriptive perspective of biodiversity, I believe that further exploration is needed to evaluate the reasons behind the results.

Spatial sampling bias:

The manuscript integrates data from literature, herbaria, and field collections; however, the potential spatial bias of sampling effort is not evaluated. It is recommended to include a sampling density map to visualize the spatial distribution of records and to explicitly discuss how such bias could influence observed richness and endemism patterns. Additionally, the use of spatial rarefaction methods or effort-weighted corrections could improve the interpretation of results.

Details of sampling design:

Although field collections are mentioned, the manuscript does not provide information on sampling effort, the number of sites, or the spatial design employed. Providing these details would allow assessment of the representativeness of records and their potential influence on spatial patterns.

Data quality and geographic precision:

There is no evaluation of geographic precision or data quality. The manuscript does not describe how duplicate records, uncertain georeferences, or historical localities were handled. Ensuring data quality is essential to confirm that the generated maps reflect biological patterns rather than artifacts of incomplete or imprecise data.

Interpretation of maps and absence of records:

The manuscript assumes that the generated maps reflect biological patterns; however, it does not discuss whether absence of records corresponds to true species absence. It is suggested to clarify that some areas with no records may result from limited sampling and to consider the potential effect of this on interpreting richness and endemism hotspots.

Sensitivity to spatial scale:

The manuscript states that the 30′ × 30′ resolution provides reliable results, but no spatial sensitivity analysis is presented. It is recommended to recalculate richness, WE, and CWE metrics at different resolutions (e.g., 15′ and 60′) and compare patterns using rank correlations and hotspot overlap. This would allow evaluation of whether the identified richness and endemism centers are robust or dependent on the selected scale.

Conclusions

The article lacks proposals for the management, conservation, and restoration of vegetation and species in the study area. I believe that this document could serve as the basis for a series of public policies for the protection of the ecosystem.

6. PLOS authors have the option to publish the peer review history of their article (what does this mean?). If published, this will include your full peer review and any attached files.

Reviewer #1: **Yes:** Eduardo Ruiz-Sanchez

Reviewer #2: No

To ensure your figures meet our technical requirements, please review our figure guidelines: s://journals.plos.org/plosone/s/figures

You may also use PLOS’s free figure tool, NAAS, to help you prepare publication quality figures: s://journals.plos.org/plosone/s/figures#loc-tools-for-figure-preparation.

---

## [Author Response · Author response to Decision Letter 1]

9 Feb 2026

Rebuttal

Reviewer #1: I have reviewed your manuscript titled "Richness and biogeography of Pteridoflora in montane forests of eastern Mexico" and found it to be well-written. The methods are appropriate, and the results and discussion are well presented. I have only two concerns:

1. You need to explicitly explain how you obtained the geographical coordinates for the specimens that lack them.

Author's response: For herbarium specimens without geographic coordinates, we used topographic maps at scale 1:50,000 produced by the Instituto Nacional de Estadística y Geografía (INEGI) and the previously published nomenclator of localities for the Sierra Madre Oriental (Ortiz-Bermúdez, 2004) to obtain the corresponding geographic coordinates. At the end, we tried to include only georeferenced and verified data in our study.

Ortiz-Bermúdez, E. (2004). Nomenclator de localidades. In: Biodiversidad de la Sierra Madre Oriental, Luna-Vega I., J.J. Morrone & D. Espinosa (eds.). pp. 25-62. Mexico City: UNAM-CONABIO.

2. The dataset used for all the analyses (species with their geographical coordinates) is not available. I encourage you to make this information public.

Author's response: the database is available through personal communication to the first author.

Reviewer #2: Methodologically, I think the article is well presented.

Although it is presented from a descriptive perspective of biodiversity, I believe that further exploration is needed to evaluate the reasons behind the results.

Spatial sampling bias:

1.The manuscript integrates data from literature, herbaria, and field collections; however, the potential spatial bias of sampling effort is not evaluated. It is recommended to include a sampling density map to visualize the spatial distribution of records.

Author's response: We have already created a sampling density map to show the spatial distribution of records. We chose not to include it in the first draft because of the manuscript's length. In this new version, we are adding the suggested map to Figure 2.

Reviewer 2: The referee suggested that the use of spatial rarefaction methods or effort-weighted corrections could improve the interpretation of results.

Author’s response: As suggested by the referee, we undertook an effort-weighted sampling method for our manuscript. In our study using the Chao1 richness estimator, we identified 567 species. According to the estimator, the average estimated richness at this sampling level was about 670 species, with a 95% confidence interval of approximately 632-729 species. The difference between the lower and upper confidence limits is roughly 100 species, indicating moderate uncertainty around the estimate, as expected in communities with many rare or low-detectability taxa.

Comparing the observed richness (567 species) to the Chao1 mean estimate (≈670 species), our data falls short by about 100 species from the estimated total richness. This indicates that our sampling completeness is approximately 85%, which is fairly high, because Chao1 uses the number of very infrequently observed species to infer unseen diversity, it tends to produce higher richness estimates when rare species are relatively common in the sample (Chao, 1984; Chao, 1987).

Overall, these results suggest that although the sampling did not fully capture the total estimated richness, it still documented a significant portion of the community. To finish the task, we decided to create a new figure to compare our results with the Chao1 richness estimator (Fig. 6).

2. Details of sampling design:

Although field collections are mentioned, the manuscript does not provide information on sampling effort, the number of sites, or the spatial design employed. Providing these details would allow assessment of the representativeness of records and their potential influence on spatial patterns.

Author's response: Our study primarily relies on herbarium data and databases, so our field collection was minimal. Our database has been compiled and curated continuously by a Mexican specialist in ferns and lycophytes for over three decades. Therefore, we can confidently state that we have sufficient information on the spatial patterns of these taxa (see paragraphs above).

3. Data quality and geographic precision:

There is no evaluation of geographic precision or data quality. The manuscript does not describe how duplicate records, uncertain georeferences, or historical localities were handled. Ensuring data quality is essential to confirm that the generated maps reflect biological patterns rather than artifacts of incomplete or imprecise data.

Author's response: For vascular plants, species occurrence data typically come from direct field observations or natural history specimens in herbaria (Daru, 2025), although specimens trump observations in representing global plant biogeographic patterns (Daru & Rodriguez, 2023). In general, no hard-and-fast rule exists for judging data quality in biogeographical analyses (Zizka et al. 2019). Indeed, collecting efforts are not randomly or regularly distributed, and biodiversity patterns are scale-dependent and sensitive to spatial resolution (James et al. 2018). Also, sampling designs were rarely quantified (Daru et al. 2018). In species occurrence records, problems with geographic location are a significant concern; particularly, erroneous or overly imprecise coordinates can bias biodiversity patterns at multiple spatial scales (Zizka et al. 2019).

As a result, data quality in online databases is a significant concern and has restricted their usefulness and reliability for research and conservation (Anderson et al. 2016). Herbarium specimens and their data have advantages that are verifiable, repeatable, sustainable, and persistent (James et al. 2018). In our paper, fortunately, most records in our database come from herbarium specimens revised directly by some of the authors, mainly the first author, a Mexican specialist in ferns and lycophytes. However, botanists have identified multiple biases in plant sampling from herbarium collections. For example, various analyses based on millions of herbarium and museum records have found spatial, temporal, taxonomic, functional trait, phylogenetic, and collector biases (Daru et al. 2018; Daru & Rodriguez 2023; Daru 2025). Limitations in our database were addressed and revised to minimize, primarily, spatial bias.

In our database, as is common with herbarium specimens, duplicate samples are found across different botanical collections. In our study, we removed duplicate records from the biogeographic analysis to ensure unique locations. Additionally, we identified and corrected records with switched latitude and longitude. We also excluded specimens with vague localities and inconsistencies. For herbarium specimens lacking geographic coordinates, we used topographic maps at a 1:50,000 scale from the Instituto Nacional de Estadística y Geografía (INEGI) and the previously published locality list for the Sierra Madre Oriental (Ortiz-Bermúdez, 2004) to obtain the corresponding coordinates. Ultimately, we aimed to include only georeferenced and verified data in our study.

Specimens stored in Natural History collections are available for long periods, sometimes up to 100 years, as in our study, because we have specimens collected in the 1880s. Unfortunately, collecting efforts are not evenly distributed over time and have declined since the mid-20th century (Daru et al. 2018; James et al. 2018). However, working with historical data presents challenges if there is no information about the species' locality or date. Additionally, the spatial location of the find may only be roughly known, or there could be issues with taxonomic naming (Lütolf et al. 2006). We examined all these issues in the historical records of ferns and lycophytes from the SMOR, and because of this, we did not include the 31 historical records from the 1800s.

Lütolf et al. (2006) highlight the use and importance of historical records, named 'ghost of past presence data', to support presence data in biogeographic studies. These records can be used as a strategy to verify a species' presence and reconfirm its presence where it was once observed. Also, these historical records are important because they can reveal changes in species distributions driven by anthropogenic activities or climate change (James et al., 2018; Franklin, 2023).

Anderson et al. (2016). Final report of the task group of GBIF data fitness for use in distribution modelling – Are species occurrence data in global online repositories fit for modelling species distributions? The case of the Global Biodiversity Information Facility (GBIF).

Daru, B. H. (2025). Tracking hidden dimensions of plant biogeography from herbaria. New Phytologist, 246(1), 61-77.

Daru, B. H., Rodriguez, J. (2023). Mass production of unvouchered records fails to represent global biodiversity patterns. Nature Ecology & Evolution, 7, 816-831.

Daru, B. H. et al. (2018). Widespread sampling biases in herbaria revealed from large-scale digitization. New Phytologist, 217, 939-955.

Franklin, J. (2023). Species distribution modelling supports the study of past, present and future biogeographies. Journal of Biogeography, 50, 1533-1545.

James et al. (2018). Herbarium data: Global biodiversity and societal botanical needs for novel research. Applications in Plant Science, 6(2), e1024.

Lütolf et al. (2006). The ghost of past species occurrence: improving species distribution models for presence-only data. Journal of Applied Ecology, 43, 802-815.

Ortiz-Bermúdez, E. (2004). Nomenclátor de localidades. In: Biodiversidad de la Sierra Madre Oriental, Luna-Vega I., J.J. Morrone & D. Espinosa (eds.). pp. 25-62. Mexico City: UNAM-CONABIO.

Zizka et al. (2019). CoordinateCleaner: Standardized cleaning of occurrence records from biological collections databases. Methods in Ecology and Evolution, 10, 744-751.

4. Interpretation of maps and absence of records:

The manuscript assumes that the generated maps reflect biological patterns; however, it does not discuss whether absence of records corresponds to true species absence. It is suggested to clarify that some areas with no records may result from limited sampling and to consider the potential effect of this on interpreting richness and endemism hotspots.

Author’s response: Fortunately, the first author of this study has been working with these organisms for more than 30 years. However, Dr. Tejero has not collected samples from all sites, so in some areas, the sampling of certain species may be incomplete.

'Presence-only' data are common sources of species distribution information, available through natural history collections like museum and herbarium collections (Lütolf et al. 2006). Absence data are rare or not easily accessible within herbarium datasets, and the absence of collections of a taxon at a specific location and time does not necessarily indicate its absence (James et al. 2018).

Mexico is a megadiverse country and, among ferns and lycophytes, is recognized as one of the Latin American countries with the highest diversity of these two plant groups (Almeida and Salino 2016). Large parts of the Neotropics remain inadequately sampled for vascular plants. The collections in this region are unevenly distributed, and many areas lack records of ferns and lycophytes (Almeida and Salino 2016). Mexico is no exception, and several regions, including the Sierra Madre Oriental, are still poorly sampled for these plants (Contreras-Medina et al. 2022). In our paper, we acknowledge this situation and state that "Some areas within the SMOR boundaries lack records of ferns or lycophytes, and 13 grid cells contain only one species. These low-diversity areas may be related to unexplored zones and insufficient sampling, as observed in many sectors of Mexico [68, 66] and the Neotropics [1, 104]."

5. Sensitivity to spatial scale:

The manuscript states that the 30′ × 30′ resolution provides reliable results, but no spatial sensitivity analysis is presented. It is recommended to recalculate richness, WE, and CWE metrics at different resolutions (e.g., 15′ and 60′) and compare patterns using rank correlations and hotspot overlap. This would allow evaluation of whether the identified richness and endemism centers are robust or dependent on the selected scale.

Author’s response: In the original version, we explained the reason for choosing the 30 × 30 minutes latitude/longitude grid-cells (see page 9). “The spatial size of grid cells allows for a reliable spatial resolution of the distribution compared with other studies conducted with plants in the SMOR [18, 20, 27] regarding biodiversity data and biological conservation. We selected this scale because similar scales have been tested in previous studies on biogeography and the diversity of various biological groups in the Mexican biota [Rodríguez et al. 2018, Villaseñor et al, 2021], including ferns and lycophytes [Contreras-Medina et al., 2022].”

A 1° × 1° grid cell spatial scale has been previously used in biogeographic analyses in the SMOR, such as in the studies by Santa Anna del Conde et al. (2009) with Cactaceae and Villaseñor & Ortiz (2022) with vascular plants. We chose a finer spatial scale to obtain more precise patterns of fern and lycophyte distribution. In biogeographic analyses at the Mexican state level, a finer scale than ours has been applied in Hidalgo (Villaseñor et al. 2022) and Oaxaca (Contreras-Medina et al. 2022), using 15 × 15-minute and 20 × 20-minute grid cells, respectively.

We consider it unnecessary to recalculate the grid-cell richness at 15 and 60 degrees, nor the endemism centers. During the development of this study, we tested different grid cell sizes, as the referee suggested, but the 30x30 size seems most suitable because many spatial studies with other Mexican taxa use this size, which is appropriate for comparative results [Rodríguez et al. 2018, Villaseñor et al., 2021].

Villaseñor JL, Ortiz E. (2022). A phytogeographic assessment of the Sierra Madre Oriental physiographic province, Mexico. Botanical Sciences, 100(4), 1102–23.

Villaseñor JL, Ortiz E, Sánchez-González A. (2022). Riqueza y distribución de la flora vascular del estado de Hidalgo, México. Revista Mexicana de Biodiversidad, 93, e933920.

Santa Anna del Conde H, Contreras-Medina R, Luna-Vega I. (2009). Biogeographic analysis of endemic cacti of the Sierra Madre Oriental, Mexico. Biological Journal of the Linnean Society, 97(2), 373–89.

6. Conclusions

The article lacks proposals for the management, conservation, and restoration of vegetation and species in the study area. I believe that this document could serve as the basis for a series of public policies for the protection of the ecosystem.

Author’s response: Indeed, we do not propose in this manuscript ideas about the management, conservation, and restoration of the vegetation of this mountain range. This task involves a different methodology than the one we are not using here. Our current objective is to know the spatial distribution of the taxa. Despite this, we have elaborated some maps of Mexican Natural protected areas that will be part of a subsequent study.

---

## [Decision Letter · Decision Letter 1]

27 Mar 2026

Richness and biogeography of Pteridoflora in montane forests of eastern Mexico

PONE-D-25-44575R1

Dear Dra Isolda Luna-Vega,

We’re pleased to inform you that your manuscript has been judged scientifically suitable for publication and will be formally accepted for publication once it meets all outstanding technical requirements.

Kind regards,

Marcela Pagano, Ph.D, M.D.

Academic Editor

PLOS One

Additional Editor Comments (optional):

Reviewers' comments:

Reviewer's Responses to Questions

**Comments to the Author**

1. If the authors have adequately addressed your comments raised in a previous round of review and you feel that this manuscript is now acceptable for publication, you may indicate that here to bypass the “Comments to the Author” section, enter your conflict of interest statement in the “Confidential to Editor” section, and submit your "Accept" recommendation.

Reviewer #3: (No Response)

Reviewer #4: All comments have been addressed

2. Is the manuscript technically sound, and do the data support the conclusions?

Reviewer #3: Yes

Reviewer #4: (No Response)

3. Has the statistical analysis been performed appropriately and rigorously? 

Reviewer #3: Yes

Reviewer #4: (No Response)

4. Have the authors made all data underlying the findings in their manuscript fully available?

Reviewer #3: Yes

Reviewer #4: (No Response)

5. Is the manuscript presented in an intelligible fashion and written in standard English?

Reviewer #3: Yes

Reviewer #4: (No Response)

6. Review Comments to the Author

Reviewer #3: In general, the study design to gather distributional data, analyses of biogeography and diversity as well as the use of statistics seem to be well developed, while the text is well organized and mostly written clearly. However, there are several points throughout the manuscript that remain unclear, which should be revised or updated and explained/discussed in more detail (see comments made within the document). For example, in the Methods section there are two floristic studies on ferns from Veracruz State that are not considered, but both include additional records that should still be included. Besides, the titles of figures and tables need some modifications, while it would be helpful to add in the Appendix another column to include the elevational range for each species. Also, the names and an explanation for the excluded species should be provided in a second Excel sheet of the Appendix. Furthermore, in the discussion the notable result of the observed hump-shaped elevational pattern with a peak at mid elevations due to the high diversity of epiphytic ferns should be explained in more detail. Finally, some statements require adequate references, while a few relevant studies should be still considered to strengthen the content and frame the paper in a wider context.

Reviewer #4: (No Response)

7. PLOS authors have the option to publish the peer review history of their article (what does this mean?). If published, this will include your full peer review and any attached files.

Reviewer #3: No

Reviewer #4: **Yes:** Laura Yáñez-Espinosa

---

## [Editor Report · Acceptance letter]

PONE-D-25-44575R1

PLOS One

Dear Dr. Luna-Vega,

I'm pleased to inform you that your manuscript has been deemed suitable for publication in PLOS One. Congratulations! Your manuscript is now being handed over to our production team.

Kind regards,

on behalf of

Dr. Marcela Pagano

Academic Editor

PLOS One